



# Producing realistic climate data with GANs

Camille Besombes[1,4], Olivier Pannekoucke[2], Corentin Lapeyre[1], Benjamin Sanderson[1], and
Olivier Thual[1,3]

[1]CERFACS, Toulouse, France
[2]CNRM, Université de Toulouse, Météo-France, CNRS, Toulouse, France
[3]Institut de Mécanique des Fluides de Toulouse (IMFT), Université de Toulouse, CNRS, Toulouse, France
[4]Institut National Polytechnique de Toulouse

**Correspondence:** besombes@cerfacs.fr, opannekoucke@cerfacs.fr, lapeyre@cerfacs.fr, sanderson@cerfacs.fr,
thual@cerfacs.fr

**Abstract.** This paper investigates the potential of a Wasserstein Generative Adversarial Networks to produce realistic weather situations when trained from the climate of a general circulation model (GCM). To do so, a convolutional neural network architecture is proposed for the generator and trained on a synthetic climate database, computed using a simple 3 dimensional climate model : PLASIM.

The generator transforms a "latent space", defined by a 64 dimensional Gaussian distribution, into spatially defined anomalies on the same output grid as PLASIM. The analysis of the statistics in the leading empirical orthogonal functions shows that the generator is able to reproduce many aspects of the multivariate distribution of the synthetic climate. Moreover, generated states reproduce the leading geostrophic balance present in the atmosphere.

The ability to represent the climate state in a compact, dense and potentially nonlinear latent space opens new perspectives
in the analysis and the handling of the climate. This contribution discusses the exploration of the extremes close to a given state and how to connect two realistic weather situations with this approach.

## 1   Introduction

The ability to generate realistic weather situations has numerous potential applications. Weather generators can be used to characterize the spatio-temporal complexity of phenomena in order for example to assess the socio-economical impact of the
weather (Wilks and Wilby, 1999; Peleg et al., 2018). However, in numerical weather prediction the dimension of a simulation can be very large, an order of $10^9$ is often encountered (Houtekamer and Zhang, 2016). The small size of ensembles used in data assimilation, due to computational limitations, leads to a misrepresentation of the balance present in the atmosphere such as an increment in the geopotential height, resulting in an unbalanced incremented wind because of localization (Lorenc, 2003). Issues of small finite samples of weather forecast ensembles could be addressed by considering larger synthetic ensembles of
generated situations. With current methods it is difficult to generate a realistic climate state at a low computational cost. This is usually done using analogs or by running a global climate model for a given time (Beusch et al., 2020) but remains costly. Generators can also be used for super-resolution so as to increase the resolution of a forecast leading to better results than interpolations (Li and Heap, 2014; Zhang et al., 2012).





The last decade has seen new kinds of generative methods from the machine learning field using artificial neural networks (NNs). Among these, generative adversarial networks (GANs) (Goodfellow et al., 2014), and more precisely Wasserstein GANs (Arjovsky et al., 2017), are effective data-driven approaches to parametrizing complex distributions. GANs have proven their power in unsupervised learning by generating high quality images from complex distributions. Gulrajani et al. (2017) trained a WGAN on the ImageNet database (Russakovsky et al., 2015), which contains over 14 million images with 1000 classes, and successfully learned to produce new realistic images. Several techniques developed for computer vision with GANs seem promising for domains in the geosciences. Notable examples of usage to date include Yeh et al. (2017) to do inpainting, where the objective is to recover a full image from an incomplete one, Ledig et al. (2017) to do super resolution, or Isola et al. (2017) to do image-to-image translation, where an image is generated from another one *e.g.* translate an image that contains a horse into one with a zebra. While there is a growing interest for using deep learning methods in weather impact or weather prediction (Reichstein et al., 2019; Dramsch, 2020), few applications have been described using GANs applied to physical fields in recent years (Wu et al., 2020). Notable examples include application to subgrid processes (Leinonen et al., 2019), to simplified models such as Lorenz '96 model (Gagne et al., 2020) or data processing like satellite images (Requena-Mesa et al., 2018). In particular, little is known about the feasibility of designing and training a generator that would be able to produce multivariate states of a global atmosphere. A first difficulty is to propose an architecture for the generator, with the specific challenge of handling the spherical geometry. Most of the neural network architectures in computer vision handle regular 2D images instead of images representing projected spherical images. Boundary conditions of these projections are not simple, as the spherical geometry also influences the spread of the meteorological object as a function of its latitude. These effects have to be considered in the neural network architecture. Another difficulty is to validate the climate resulting from the generator compared with the true climate.

In this study, in order to evaluate the potential of GANs applied to the global atmosphere, a synthetic climate is computed using the PLASIM global circulation simulator (Fraedrich et al., 2005), a simplified but realistic implementation of the primitive equations on the sphere. An architecture is proposed for the generator and trained using an approach based on the Wasserstein distance. A multivariate state is obtained by the transformation of a sample from a Gaussian random distribution in 64 dimensions by the generator. Thanks to this sampling strategy, it is possible to compute a climate as represented from the generator. Different metrics are considered to compare the climate of the generator with the true climate, and to assess the realism of the generated states. Because the distribution is known, the generator provides a new way to explore the climate *e.g.* simulating the intensification of a weather situation or interpolating two weather situations in a physically plausible manner.

The article is organized as follows. The formalism of WGAN is first introduced in Section 2, with the details of the proposed architecture. Then, Section 3 evaluates the ability of the generator to reproduce the climate of PLASIM with assessment of the climate states that are produced by the generator. The conclusions and perspectives are given in Section 4.



## 2 Wasserstein Generative Adversarial Network to characterize the climate

### 2.1 Parametrizing the climate of the Earth system

The Earth system is considered as being the solution of an evolution equation

$$\partial_t \chi = \mathcal{M}(\chi), \tag{1}$$

where $\chi$ denotes the state of the system at a given time and $\mathcal{M}$ characterizes the dynamics including the forcing terms *e.g.* the solar annual cycle. While $\chi$ should stand for continuous multivariate fields, we consider its discretization in a finite grid so that $\chi \in X$ with $X = \mathbb{R}^n$, where $n$ denotes the dimension. Equation 1 describes a chaotic system. The *climate* is the set of states of the system along its time evolution. It is characterized by a distribution or a probability measure, denoted $p_{\text{clim}}$.

Obtaining a complete description of $p_{\text{clim}}$ is intractable due to the complexity of natural weather dynamics, and because a climate database, $p_{\text{data}}$, is limited by numerical resources and is only a proxy for this distribution.

For instance, in the present study, the true weather dynamics $\mathcal{M}$ are replaced by the PLASIM model that has been time integrated over 100 years of 6h forecasts. Accounting for the spin up, the first ten years of simulation are ignored. Thus, the climate $p_{\text{clim}}$ is approximated from the resulting climate database of 90 years, $p_{\text{data}}$. The synthetic dataset is presented in detail in Section 3.1.

Thus, $p_{\text{data}}$ lives in the $n$-dimensional space $X$, but it is non-zero only on an $m$-manifold $\mathbb{M}$ (where $m \ll n$) that can be fractal. The objective is to learn a mapping

$$g : Z \mapsto X, \tag{2}$$

from $Z = \mathbb{R}^m$, the so-called latent space, to $X$. Moreover, $g$ must transform a Gaussian $\mathcal{N}(0, \mathbf{I}_m)$ to $p_{\text{data}} \subset \mathbb{M}$

The main advantage of such a formulation is to have a function $g$ that maps a low dimensional continuous space $Z$ to $\mathbb{M}$. This property could be useful in the domain of geoscience notably in climate sciences where a high dimensional space is ruled by important physical constraints and parameters.

Here the generator is a good candidate to learn the physical constraints that make a climate state realistic without the need to run a complete simulation. The construction of the generator is now detailed.

### 2.2 Background on Wasserstein generative adversarial networks

To characterize the climate, we first introduce a simple Gaussian distribution $p_z = \mathcal{N}(0, \mathbf{I}_m)$ of zero mean and covariance the identity matrix $\mathbf{I}_m$, defined on the space $Z = \mathbb{R}^m$, called the latent space. The objective of an adversarial network is to find a non-linear transformation of this space $Z$ to $X$ as written in Eq. (2) so that the Gaussian distribution maps to the climate distribution *i.e.* $g_\#(p_z) = p_{\text{clim}}$ where $g_\#$ denotes the pushforward of a measure by the map $g$, defined here as follows: for any measurable set $E$ of $X$, $g_\#(p_z)(E) = p_z(g^{-1}(E))$ where $g^{-1}(E)$ denotes the measurable set of $Z$ that is the pre-image of $E$ by $g$. The latent space, $Z$, can be seen as an encoded climate space where each point drawn from $p_z$ corresponds to a realistic climate state and where the generator is the decoder. Looking for such a transformation is non-trivial.





The search is limited to a family of transformations $\{g_\theta\}$, characterized by a set of parameters $\theta$. Thus, for each $\theta$, the non-linear transform of the Gaussian $p_z$ by $g_\theta$ is a distribution $p_\theta$. The goal is then to find the best set of parameters, $\theta^*$ such that $\theta^* = \text{argmin}_\theta \; di(p_\theta, p_{\text{clim}})$ where $di$ is a measure of the discrepancy between the two distributions, so that $p_{\theta^*}$ approximates $p_{\text{clim}}$. This method is known as the generative learning, where $g_\theta$ is implemented as a neural network of trainable parameters

$\theta$. Note that, being a neural network, the resulting $g_\theta$ is then a differentiable function.

Even with this simplified framework, the search of an optimal $\theta$ is not easy. One of the difficulties is that the differentiability of $g_\theta$ requires the comparison of continuous distribution $p_\theta$ with $p_{\text{clim}}$, which is not necessarily a density on a continuous set. To alleviate this issue, Arjovsky et al. (2017) introduced an optimization process based on the Wasserstein distance defined for the two distributions $p_{\text{clim}}$ and $p_\theta$ by

$$W(p_\theta, p_{\text{clim}}) = \inf_{\gamma \in \Pi(p_\theta, p_{\text{clim}})} \mathbb{E}_{(x,y)} \left[ \|x - y\| \right], \tag{3}$$

where $\Pi(p_\theta, p_{\text{clim}})$ denotes the set of all joint distributions $\gamma(x, y)$ whose marginals are respectively $\int_y \gamma(\cdot, dy) = p_\theta$ and $\int_x \gamma(dx, \cdot) = p_{\text{clim}}$. The Wasserstein distance, also called Earth mover distance (EMD), comes from optimal transport theory and can be seen as the minimum work required (in the sense of $mass \times transport$) to transform the distribution $p_\theta$ into the distribution $p_{\text{clim}}$. Thus, the set $\Pi(p_\theta, p_{\text{clim}})$ can be seen as all the possible mappings, also called couplings, to transport the

mass from $p_\theta$ to $p_{\text{clim}}$. The Wasserstein distance is a *weak distance* : it is based on the expectation, which can be estimated whatever the kind of distributions. Hence, the optimization problem states as

$$\theta^* = \text{argmin}_\theta W(p_\theta, p_{\text{clim}}), \tag{4}$$

which leads to the Wasserstein GAN (WGAN) approach.

One of the major advantages of the Wasserstein distance is that it is real-valued for non-overlapping distributions. Indeed, the

Kullback-Leibler (KL) divergence is infinite for disjoint distributions and using it as a loss function leads to vanishing gradient Arjovsky et al. (2017). The Wasserstein Distance does not exhibit vanishing gradients when distributions do not overlap, as did the KL divergence in the original GAN formulation.

Unfortunately, the formulation in Eq. (3) is intractable. A reformulation is necessary using the dual form discovered by Kantorovich (Kantorovich and Rubinshtein, 1958). Reframing the problem as a linear programming problem yields

$$W(p_\theta, p_{\text{clim}}) =$$

$$\sup_{f \in 1-\text{Lipshitzian}} \left[ \mathbb{E}_{x \sim p_{\text{clim}}} \left[ f(x) \right] - \mathbb{E}_{x \sim p_\theta} \left[ f(x) \right] \right], \tag{5}$$

where $1-\text{Lipshitzian}$ denotes the set of Lipshitzian functions $f : \mathbb{R}^n \to \mathbb{R}$ of coefficient 1 *i.e.* for any $(x_1, x_2) \in \mathbb{R}^n$, $|f(x_1) - f(x_2)| \leq \|x_1 - x_2\|$, $\|\cdot\|$ being the Euclidian norm of $\mathbb{R}^n$. For any $1-\text{Lipshitzian}$ function $f$ the computation of Eq. (5) is

simple: the first expectation can be approximated by :

$$\mathbb{E}_{x \sim p_{\text{clim}}} \left[ f(x) \right] \approx \mathbb{E}_{x \sim p_{\text{data}}} \left[ f(x) \right], \tag{6}$$




where the right-hand side is computed as the empirical mean over the climate database $p_{\text{data}}$ that approximates $p_{\text{clim}}$ in the weak sens Eq. (6). The second expectation can be computed from the equality

$$\mathbb{E}_{x\sim p_\theta}\left[f(x)\right] = \mathbb{E}_{z\sim\mathcal{N}(0;\mathbf{I}_m)}\left[f(g_\theta(z))\right], \tag{7}$$

120 where the expectation of the right-hand side can be approximated by the empirical mean computed from an ensemble of samples of $z$ which are easy to sample due to the Gaussianity.

However, there is no simple way to characterize the set of $1-$ Lipshitzian functions which limits the search of an optimal function in Eq. (5). Instead of looking at all $1-$ Lipshitzian functions, a family of functions, $\{f_w\}$ parametrized by a set of parameters $w$, is introduced. In practice, it is engendered by a neural network with trainable parameters $w$, called the *critic*.

125 Finally, if the weights of the network are constrained to a compact space $\mathcal{W}$, which can be done by the weight clipping method described in Arjovsky et al. (2017), then $\{f_w\}_{w\in\mathcal{W}}$ will be K-Lipschitzian with K depending only on $\mathcal{W}$ and not on individual weights of the network. This yields :

$$\max_{w\in\mathcal{W}}\left[\mathbb{E}_{x\sim p_{data}}\left[f_w(x)\right] - \mathbb{E}_{z\sim\mathcal{N}(0;\mathbf{I}_m)}\left[f_w(g_\theta(z))\right]\right] \leq$$
$$\sup_{f\in 1-\text{Lipshitzian}}\left[\mathbb{E}_{x\sim p_{data}}\left[f(x)\right] - \mathbb{E}_{z\sim\mathcal{N}(0;\mathbf{I}_m)}\left[f(g_\theta(z))\right]\right] \tag{8}$$

which tells us that the critic tends to the Wasserstein distance when trained optimally *i.e.* if we find the $\max$ in Eq. (8) and if $f$ is 130 in (or close to) $\{f_w\}_{w\in\mathcal{W}}$. The weight clipping method was replaced by the gradient penalty method in Gulrajani et al. (2017) because it diminished the training quality as mentioned in Arjovsky et al. (2017). Because it results from a neural network, any function $f_w$ is differentiable, so that the $1-$ Lipshitzian condition remains to ensure a gradient norm bounded by 1 *i.e.* for any $x\in X, ||\nabla f_w(x)|| \leq 1$. To do so, Gulrajani et al. (2017) have proposed to compute the optimal parameter $\tilde{w}(\theta)$ as the solution of the optimization problem

135 $$\tilde{w}(\theta) = \text{argsup}_w L(\theta, w) \tag{9}$$

where $L$ is the cost function

$$L(\theta, w) = \mathbb{E}_{x\sim p_{data}}\left[f_w(x)\right] - \mathbb{E}_{z\sim\mathcal{N}(0;\mathbf{I}_m)}\left[f_w(g_\theta(z))\right]$$
$$+ \lambda\mathbb{E}_{\hat{x}\sim\hat{p}}\left[\left(||\nabla f_w(\hat{x})|| - 1\right)^2\right] \tag{10}$$

140 with $\lambda$ the magnitude of the gradient penalty and where $\hat{x}$ is uniformly sampled from the straight line between a sample from $p_{\text{data}}$ to a sample from $p_\theta$ (line 8) of Algorithm 1. The optimal solution $\tilde{w}(\theta)$ is obtained from a sequential method where each step writes

$$w_{k+1} = w_k + \beta_k\nabla_w L(\theta, w_k), \tag{11}$$

where $\beta_k$ is the magnitude of the step. In an adversarial way, Eq. (10) could be solved sequentially *e.g.* by the steepest descent 145 algorithm with an update given by :

$$\theta_{q+1} = \theta_q - \alpha_q\nabla_\theta W(p_{\theta_q}, p_{\text{clim}}), \tag{12}$$





where $\alpha_q$ is the magnitude of the step. We chose to use the two-sided penalty for gradient penalty method, as it was shown to work well in Gulrajani et al. (2017). At convergence, the Wasserstein distance is approximated by

150 $W(p_\theta, p_{\text{clim}}) \approx$

$$\mathbb{E}_{x \sim p_{data}} \left[ f_{\tilde{w}(\theta)}(x) \right] - \mathbb{E}_{z \sim \mathcal{N}(0; \mathbf{I}_m)} \left[ f_{\tilde{w}(\theta)}(g_\theta(z)) \right]. \quad (13)$$

Hence, the solution of the optimization problem Eq. (4) is obtained from a sequential process composed of two steps, summarized in the Algorithm 1. In the first step, the weights of the generator are frozen with a given set of parameters $\theta_q$ and the critic neural network is trained in order to find the optimal parameter $\tilde{w}(\theta_q)$ solution Eq. (9) (lines 3 – 11 in 1). In the second step,

the critic is frozen and the generator is set as trainable in order to compute $\theta_{q+1}$ from Eq. (12) (lines 12 – 17 in 1). Note that in the Algorithm 1, the steepest descent is replaced by an Adam optimizer (Kingma and Ba, 2014), a particular implementation of stochastic gradient descent which has shown to be efficient in deep learning.

The following sections will aim to create a climate data generator from the WGAN method. The next section will describe the architecture of the network adapted to the complexity of the dataset used.

---

**Algorithm 1** WGAN training algorithm.

---

**Require:** Learning rate $lr$, batch size $b$, $n_{critic}$ number of iteration of the critic per generator iteration.

**Require:** $w_0$ and $\theta_0$ respectively the initial critic and generator parameters.

1:   # Optimization cycle

2:   **while** $\theta$ has not converged **do**

3:       # 1. Computation of the Wasserstein distance by maximization over $1-$Lipshitzian functions

4:       **for** $t = 0, \ldots, n_{critic}$ **do**

5:           # 1.1 Computation of the gradient for the $1-$Lip. function.

6:           Sample$\{x^{(i)}\}_{i=1}^b \sim P_{data}$ a batch from the real data.

7:           Sample$\{z^{(i)}\}_{i=1}^b \sim P_\theta$ a batch from the generated data.

8:           Sample$\{\hat{x}^{(i)}\}_{i=1}^b$ where $\hat{x} = \xi x + (1 - \xi) g_\theta(z)$ where $\xi \sim \mathcal{U}[0,1]$

9:           $grad_w \leftarrow \nabla_w \left[ \frac{1}{b}\Sigma_{i=1}^b f_w(x^{(i)}) - \frac{1}{b}\Sigma_{i=1}^b f_w(g_\theta(z^{(i)})) + \frac{\lambda}{b}\Sigma_{i=1}^b \left( ||\nabla f_w(\hat{x}^{(i)})|| - 1 \right)^2 \right]$

10:          # 1.2 Update the parameter $w$ to maximize Eq. (5)

11:          $w \leftarrow w + lr * Adam(w, grad_w)$

12:      **end for**

13:      # 2. Update the generator

14:      # 2.1 Compute the gradient of the Wasserstein distance

15:      Sample$\{z^{(i)}\}_{i=1}^b \sim P_\theta$ a batch from the generated data.

16:      $grad_\theta \leftarrow \nabla_\theta \left[ \frac{1}{b}\Sigma_{i=1}^b f_w(g_\theta(z^{(i)})) \right]$

17:      # 2.2 Update the parameter $\theta$ to minimize the Wasserstein distance

18:      $\theta \leftarrow \theta - lr * Adam(\theta, grad_\theta)$

19: **end while**

---





## 2.3 Neural network implementation

WGANs are known to be time-consuming to train, usually needing a high number of iterations due to the alternating aspect of the training algorithm between the critic and the generator. Our initial architecture used a simple convolutional network for both, with a high number of parameters, but it proved difficult to train fitting multimodal distribution such as green distributions in the left panels in Fig. 16. That is why for this study a ResNet-inspired architecture (He et al., 2016) was chosen. The goal of the Residual network is to reduce the number of parameters of the network and avoid gradient vanishing which is a recurrent problem for deep networks that results in an even slower training.

A network is composed of a stack of layers, when a specific succession of layers is used several times we can refer to it as a *block*. The link between two layers is named a connection, a shortcut connection refers to a link between two layers that are not successive in the architecture. A residual block Fig. 2, 3 is composed with stacked convolution and a parallel identity shortcut connection. The idea is that it is easier to learn the residual mapping than all of it, so residual blocks can be stacked without observing a vanishing gradient. Moreover, a residual block can be added to an N layer network without reducing its accuracy because it is easier to learn $F(x) = 0$ by setting all the weights to 0 than it is to learn the identity function. Residual blocks allow building deeper networks without loss in accuracy.

To impose the periodic boundary condition, it was necessary to create a *wrap padding layer*, which takes multiple columns at the east side and concatenates them to the west side, and vice-versa. In the critic, the wrap padding is only after the input, since the critic will discriminate the images from the generator that are not continuous on the west-east direction. In the generator, the *wrap padding layer* is in every residual block, it is necessary because the reduced size of the convolution kernel compared to the image size, makes it more difficult for the network to extract features from both sides of the image simultaneously. The North-South boundary is padded by repeating the nearest line, called *nearest padding layer*. In Fig. 1, 2, 3, 4, 5 padding layers argument have to be understood as $(longitude\ direction,\ latitude\ direction)$ where the integer means the number of columns or rows to be taken of each side and placed to the other one, *e.g. Wrappadding*$(0, 3)$ means the output image is 6 columns larger than the input. If the argument is not mentioned then the argument for wrap and nearest padding are $(0, 1)$ and $(1, 0)$ respectively.

### 2.3.1 Critic network

The critic network input has the shape of a sample from the dataset $X \in \mathbb{R}^{nlat \times nlon \times nfield}$

Its output must be a real number because it is an approximation of the Wasserstein distance between the distribution of batch of images from the dataset and the one from the generator that is being processed. The architecture ends with a dense layer of one neuron with linear activation. The core of the structure is taken from the residual network and can be seen in Fig. 1. After the custom padding layers mentioned previously the critic architecture is a classical residual network, starting with a convolution with $7 \times 7$ kernels, followed by a max pooling layer to reduce the image size and a succession of convolutional and identity blocks Fig. 2, 3. At each convolutional block, the image size is divided by a factor 2. Finally, an average pooling is done, and the output is fed to a fully connected layer of 100 neurons, then to the output neuron. Batch normalization is not



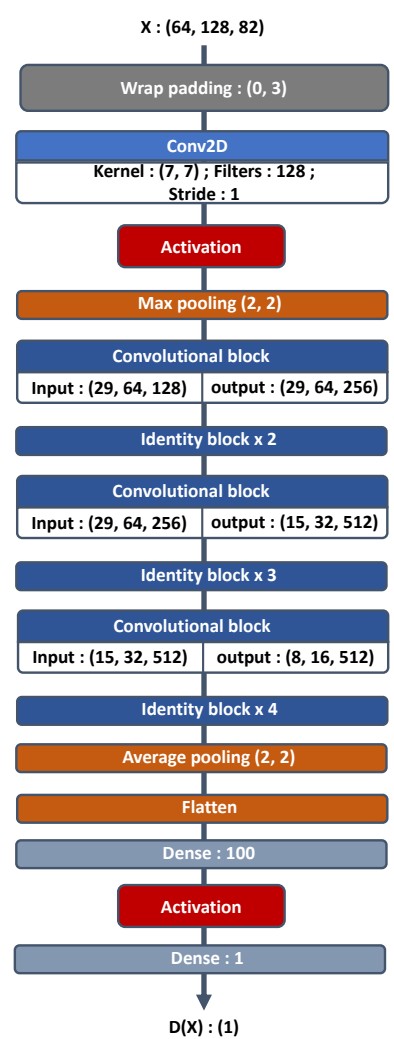

**Figure 1.** Critic architecture.

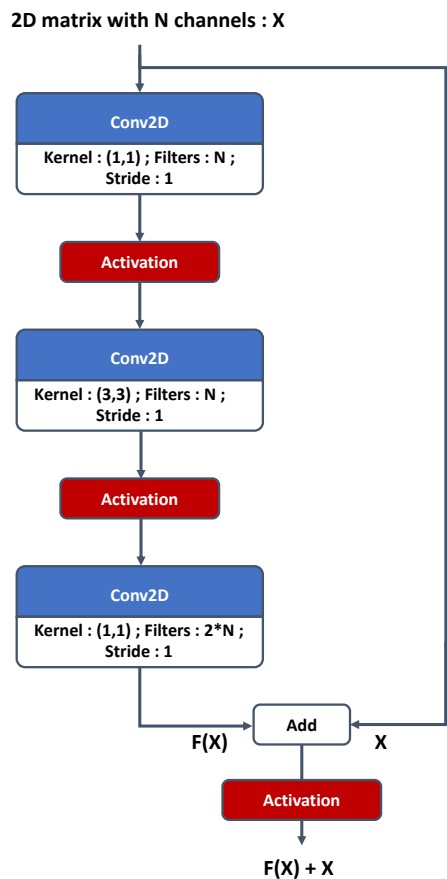

**Figure 2.** Residual blocks for the critic

present in the critic architecture following Gulrajani et al. (2017), the batch normalization changes the discriminator's problem by considering all the batch in the training objective whereas we are already penalizing the norm of the critic's gradient with respect to each sample in the batch.

### 2.3.2 Generator architecture

The input of the generator network (see Fig. 4) is an $m$-dimensional vector containing noise drawn from the normal distribution $\mathcal{N}^m(0, \mathbf{I}_m)$, for the numerical experiment $m = 64$. The output of the generator has the shape of a sample of the dataset $X \in \mathbb{R}^{nlat \times nlon \times nfield}$. The input is passed through a fully connected layer of output shape $(8, 16, 128)$ and fed to residual blocks. The rest of its architecture is also a residual network with a succession of modified convolutional blocks (relative to the one in the critic network). Modifications of the convolutional block are the following :

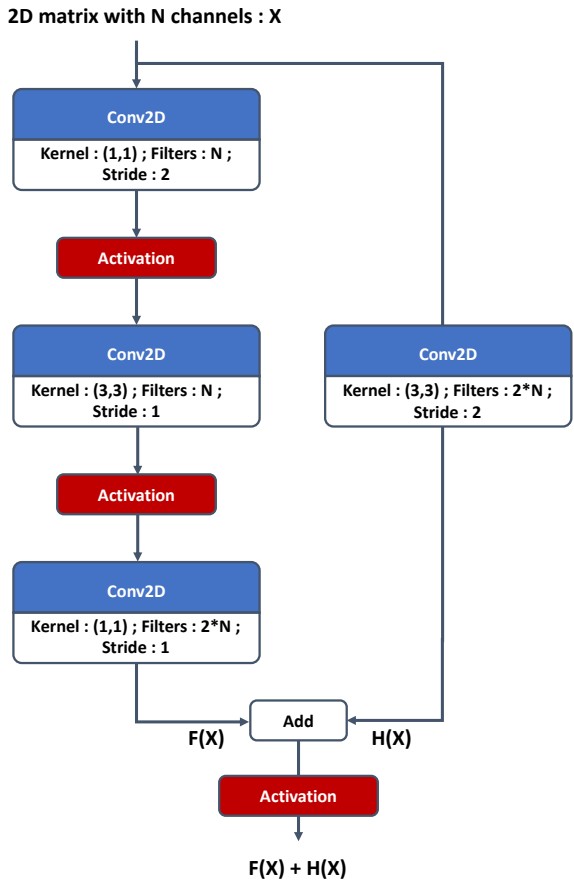

**Figure 3.** Residual blocks for the critic

1. An up sampling layer is added to increase the image size for some convolutional block.

2. Wrap and nearest padding layers are added in respectively West-East and North-South direction.

3. A batch normalization layer is present after convolutional layers.

One could argue that the ReLU activation function is not differentiable in 0 but this is managed by taking the left derivative in the software implementation. The study does not claim that the network architectures used are optimal, the computational burden was too high to run a parameter sensitivity study. Guidelines from Gulrajani et al. (2017) were followed and the hyper parameters were adapted to the current problem. It showcases an example of hyper parameters producing interesting results, and inspired readers are encouraged to modify and improve this architecture.





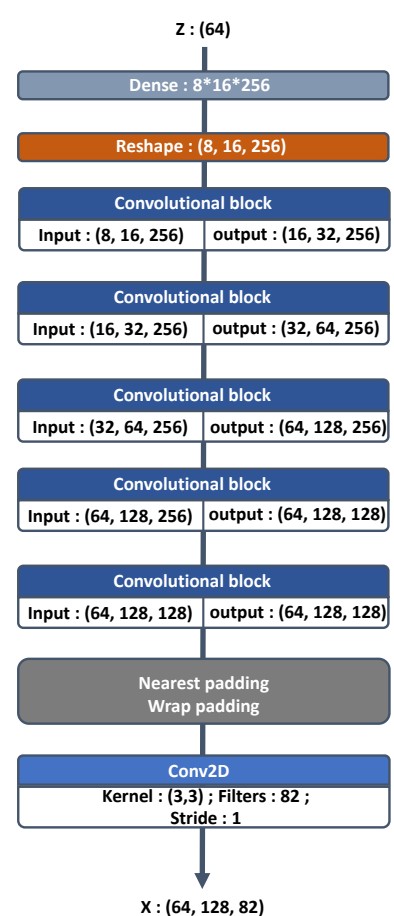

**Figure 4.** Generator architecture.

| Hyper parameters | Network | |
| --- | --- | --- |
| | Generator | Critic |
| Iterations | 30 000 | 150 000 |
| Batch Size | 128 | 128 |
| Optimizer | Adam | Adam |
| Initial learning rate (lr) | $1e^{-3}$ | $1e^{-3}$ |
| Learning rate decay every 3000 iterations | 0.9 | 0.9 |
| Number of trainable weights | $1.5e^{6}$ | $4e^{6}$ |
| $\lambda$ in Eq. (10) | | 10 |

**Table 1.** Hyper parameters for training step.


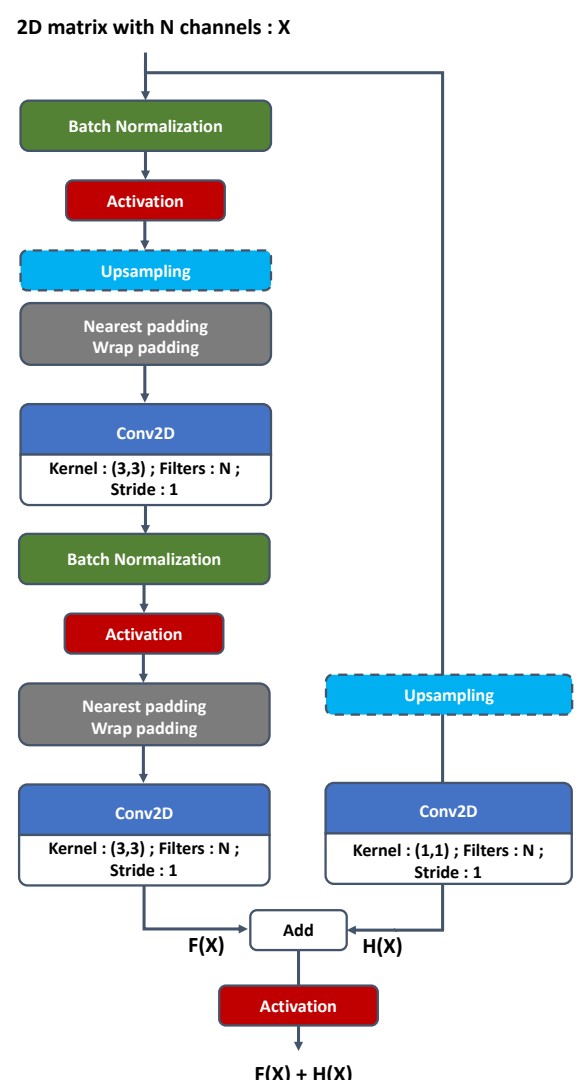

**Figure 5.** Resblock for the generator.





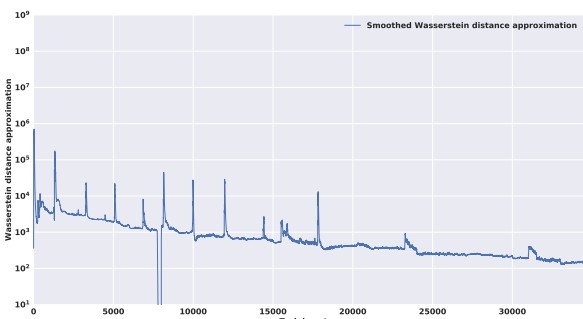

**Figure 6.** Smoothed version of the Wasserstein distance computed during the training. The vertical axis is in log scale.

### 2.3.3 Training parameters

For the training phase, the neural network's hyper parameters are summarized in the Tab. 1. The training was performed on an Nvidia Tesla V100-SXM2 of 32 GB of memory over 2 days. The choice of the optimizer, initial learning rate, weight of gradient penalty ($\lambda$ in Eq. (10)) and the ratio between critic and generator iteration were directly taken from (Gulrajani et al., 2017). The iterations mentioned in the Tab. 1 are the number of batches seen by each neural network.

The training loss in Fig. 6 was smoothed using exponential smoothing

$$s_t = \alpha y_t + (1 - \alpha) s_{t-1}, \tag{14}$$

where $y_t$ is the value of the original curve at index $t$, $s_t$ is the smoothed value at index $t$ and $\alpha$ is the smooth factor (equal to $0.9$ here). An initial spin up of the optimization process tends to exhibit an increase in the loss the first steps of the training phase before decreasing. This can be explained by the lack of useful information in the gradient due to the initial random weights in the network. A decrease of the Wasserstein distance can be seen in the Fig. 6, which indicates a convergence during the training phase. Moreover, at the end of the training a first experiment was conducted to see if the generations are present in the dataset. The histogram of the Euclidian distance divided by the number of pixels in one sample between one generation and all the dataset can be seen in Fig. 7. Here, one can see that the minimum is around $0.8$ which shows that the generated image is not inside the dataset. This experiment shows that the generator is able to generate samples without reproducing the dataset. It should be noted that in the WGAN framework, the generator never directly sees a sample from the dataset.

There is no stopping criteria for the training, and it was stopped after 35000 iterations in the interest of computational cost. It should be highlighted that the performance of generative networks and especially GANs is difficult to evaluate. In the deep learning literature, the quality of the images generated is assessed using a reference image dataset such as ImageNet (Russakovsky et al., 2015), and computing the inception score (IS) or the Fréchet Inception Distance (FID). Both use the inception network trained on ImageNet : the IS measures the quality and diversity of the images by classifying them and measuring the entropy of the classification, while the FID computes a distance between the features extracted by the inception network, and is more robust to GANs mode collapse.





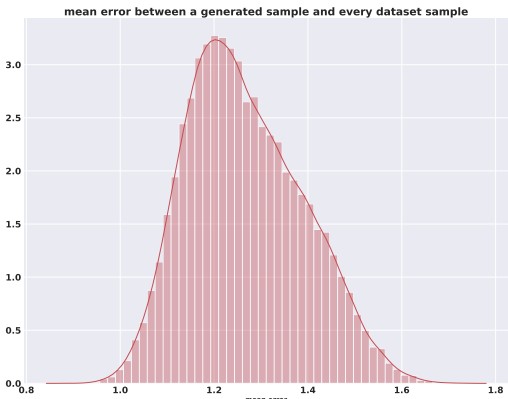

**Figure 7.** 2-norm distance between a generated sample and all the dataset sample

Because our study does not apply to the ImageNet dataset, it is necessary to compute our own metrics. Section 3 proposes an approach for this kind of method in the domain of geosciences and more precisely the study of atmospheric fields. Our main

objective is to assess the fitting quality of the dataset climate distribution.

## 3  Evaluation and exploration of the generator

The metrics by which the results will be analyzed are visual aspects, capacity to generate atmospheric balances and statistics of the generations compared to climate distribution. For the latter, the chosen metric is the Wasserstein distance. Because it is the same metric the generator has to minimize during the training step, it seems a good candidate to assess the training quality.

One could argue that the network is overly trained on this metric, that is why we use other metrics such as mean and standard deviation differences and singular value decomposition to complete our analysis. Finally, because no trivial stop criteria is available it is interesting to see where the magnitude of the Wasserstein distance is large so as to diagnose some limitations of the trained generator that would provide some ideas of improvements.

### 3.1  Description of the synthetic dataset

To create synthetic data, a climate model known as PLASIM (Fraedrich et al., 2005) was used which is a General Circulation Model (GCM) of medium complexity based on a simplified general circulation model PUMA (Portable University Model of the Atmosphere) (Fraedrich et al., 1998). This model based on primitive equations is a simplified analogue for operational Numerical Weather Prediction (NWP) models. This choice facilitates the generation of synthetic data thanks to its low resolution and reasonable computational cost. Different components can be added to the model in order to improve the circulation

simulation such as effect of ocean with sea ice, orography with biosphere or annual cycle.



| Variables | | | |
|---|---|---|---|
| Name | Short name | Prognostic | Diagnostic |
| Temperature ($K$) | $ta$ | × | |
| Eastward wind ($m.s^{-1}$) | $ua$ | | × |
| Northward wind ($Pa.s^{-1}$) | $va$ | | × |
| Relative humidity ($frac.$) | $hus$ | × | |
| Vertical velocity ($Pa.s^{-1}$) | $wap$ | | × |
| Vorticity ($s^{-1}$) | $\zeta$ | × | |
| Divergence ($s^{-1}$) | $d$ | × | |
| Geopotential height ($gpm$) | $zg$ | × | |
| ln(Surface pressure) | $P$ | | × |
| Latitude ($degrees$) | $lat$ | | × |

**Table 2.** Variables used in the dataset.

A 100-years daily simulation was run on a T42 resolution (an approximate resolution of 2.8 degrees). We used orography and annual cycle parametrization, ocean and biosphere modelisation were turned off in order to keep the dataset simple enough for our exploratory study. We removed the first 10 years in order to keep only the stationary part of the simulation. These resulting 90 years of simulation constitute the sampling of the climate distribution that we aim to reproduce. As preprocessing, each of the channels was normalized.

Each database sample is an 82 ($nfield$) channels 2D matrix of size 64 ($nlat$) by 128 ($nlon$) pixels. The channels represent 7 physical 3D variables: the temperature ($ta$), the eastward ($ua$) and northward ($va$) wind, relative humidity ($hus$), vertical velocity ($wap$), the relative vorticity ($\zeta$), divergence ($d$) and geopotential height ($zg$) on 10 pressure levels from 1000 hPa to 100 hPa ; plus the surface pressure ($ps$). Another channel was added to represent the latitude, it is an image going from -1 at the top of the image (North Pole) to 1 at the bottom (South Pole) on every column. It was found that hard coding the latitude in the data improved the learning of physical constraints, allowing the network to be sensitive to the fact that the data are represented by the equirectangular projection of the atmospheric physical fields and, for example, the size of meteorological objects increases closer to the poles. Finally, the choice of having diagnostic variables in the dataset was to help the post-processing, and assessment of their necessity requires further research.

## 3.2 Comparison between climate dataset and generated climate

Our study aims to have a generator able to reproduce the climate distribution present in the dataset made from the low resolution GCM PLASIM. This section proposes a way to assess the quality of the distribution learned by the WGAN.

Each generated sample is compared with dataset samples Fig. 8 and 9 show a sample where only the pressure levels 1000, 500 and 100 hPa are represented for readability. It should be noted that the generated fields seem to be spatially noisy compared

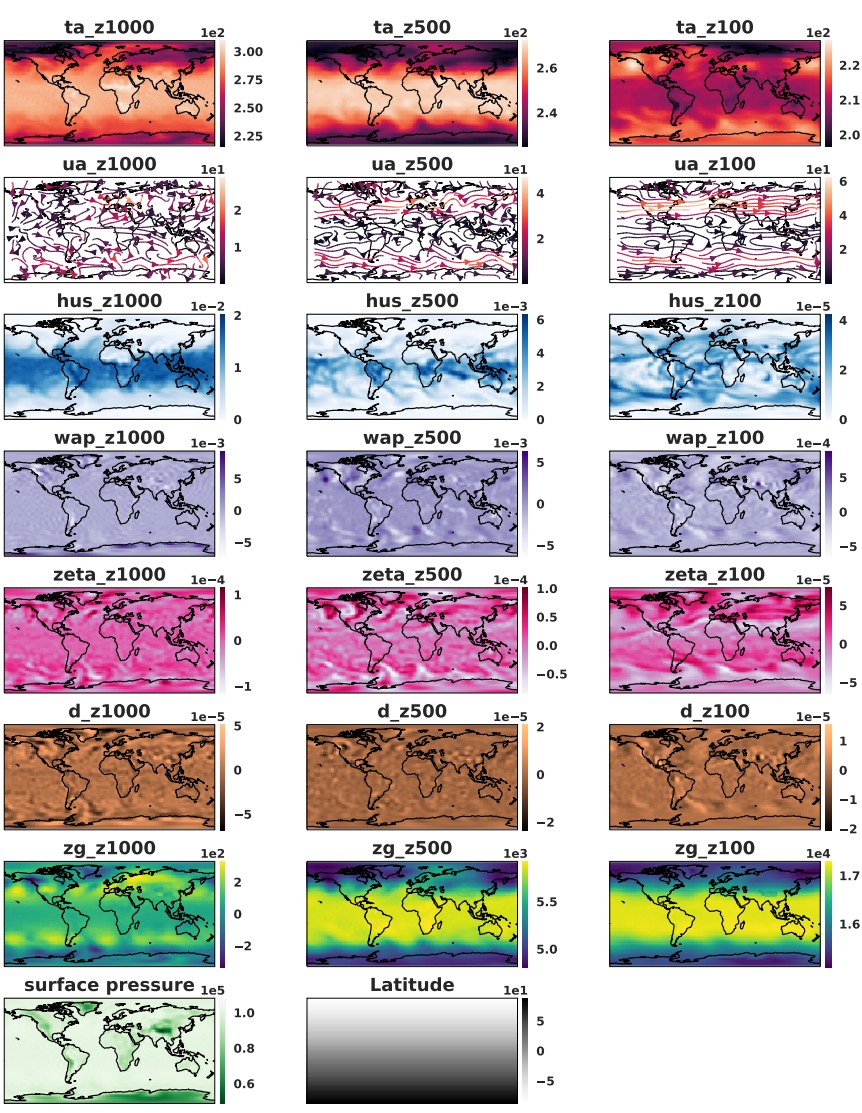

**Figure 8.** Sample on 3 different pressure levels (1000, 500 and 100 hPa) taken from the dataset. The samples were horizontally transposed in order to have a Europe at the center of the images. Coastlines were added a posteriori for readability. Units available in Tab. 2.





to the dataset. The periodic boundary is respected knowing that in the dataset the borders are located at the longitude 0° were no discontinuities can be observed. In the figures, the image is translated in order to have Europe at the center of the image and see if some discontinuities remain.

In order to quantitatively assess the generator quality, the Fig. 10 and 11 show the mean and standard deviation pixel wise differences over 10800 samples (equivalent to 30 years of data) between normalized dataset and generations. It appears that

fields where small scales patterns are present are the most difficult to fit for the generator.

In order to go further in the analysis of the generated climate states, a singular value decomposition (SVD) was performed over 30 years of the dataset (renormalized over the 30 years). Then the same number of generated data was considered and projected on the 5 first principal components of the SVD that represent 75% of explained variance of the dataset. We observed that their projections are similar in the different plans (see Fig. 12). The scatter plots shows the conservation of the 2 principal

components to a high accuracy. The distribution of the projected components are also matching which imply that the generator reproduces the basic covariance structure of the original dataset. Also, in Fig. 13 the dot product is represented between SVD components derived from the dataset $(u_i)_{i \in \{0,...,4\}}$ and another one derived from the generated data $(v_i)_{i \in \{0,...,4\}}$. Fig. 13 represents the cross-covariance matrix defined by $s_{ij} = u_i \bullet v_j$. Values close to 1 or $-1$ shows that eigenvector for both datasets (original and generated) are similar. This is another way of assessing whether the covariance structure of the original data is

being preserved. And the Fig. 13 shows that the 5 eigenvectors are similar. One should note that the SVD algorithm used from Pedregosa et al. (2011) suffers from sign indeterminacy, meaning that the sign of SVD components depend on the random state and the algorithm. For this reason, we consider the dot product both close to 1 or $-1$. One should note that an inversion remains between the components with index 3 and 4 which could be explained by a difference of eigenvalue order (sorted in decreasing order) in each dataset that determines the order of eigenvectors. The 4th principal direction (index 3 in the figure) of

the generated data represents more variation of the generated dataset than the same direction explains variation in the original dataset. Figure 14 shows clearly the inversion of the last principal components between the dataset and generations. This suggests be a way of improving our method in future work.

Figure 16 shows the temperature (at the pressure level 1000 hPa) distribution at different pixel locations corresponding to the red dots in Fig. 15. Different latitudes (42°, -2° and -70°) were chosen to represent diverse distributions. A value of Wasserstein

distance is associated with each plot, representing the distance between the two normalized distributions. It is notable that the Wasserstein distance in the context of GAN training was introduced by Arjovsky et al. (2017) in order to avoid the mode collapse phenomenon where the generated samples produced by the GAN are representing only one mode of the distribution. In Fig. 16 even if the figure shows that some bimodal distributions remain approximated by a unimodal distribution, the span of these distributions cover the multiple modes of the targeted distribution. This explains why the higher Wasserstein distance

in the figure is in the top-left panel, since despite the bimodal generated distribution the high temperature values don't seem to be represented by the generated samples.

It follows that a good way to see the general statistics learned by the generator is to plot the Wasserstein distance for every pixel and for every variable. This result can be visualized spatially in Fig. 17 where we observe that certain variables are better fitted by the generator than others. The figure shows also that areas with more variability such as land areas and more precisely

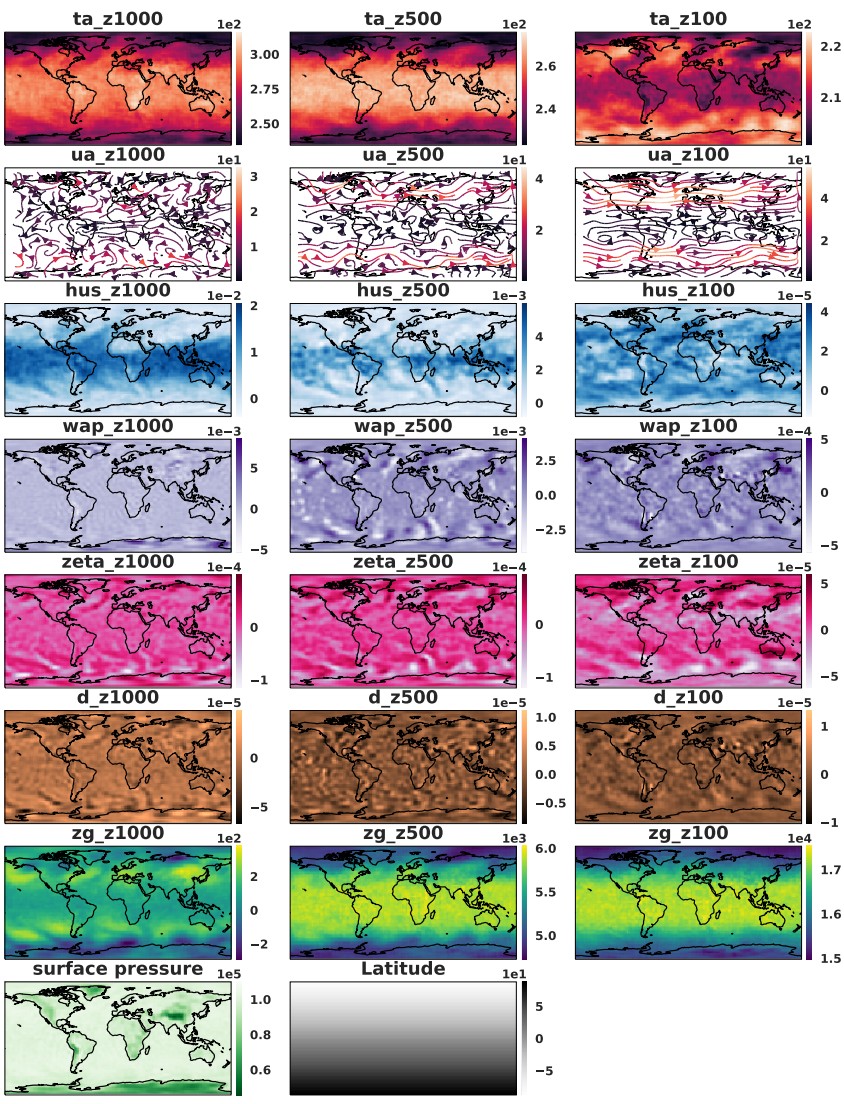

**Figure 9.** Sample on 3 different pressure levels (1000, 500 and 100 hPa) generated by the network. The samples were horizontally transposed in order to have a Europe at the center of the images to verify the quality of the periodic boundary. Coastlines were added a posteriori for readability.





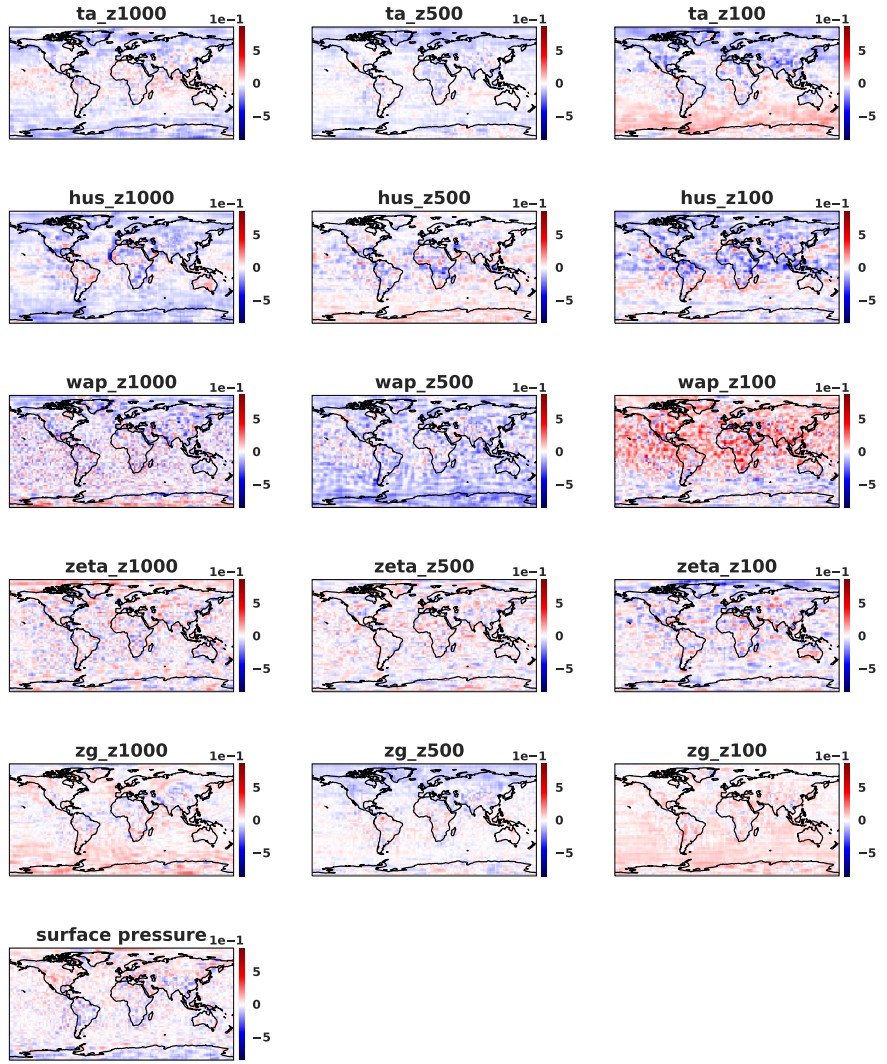

**Figure 10.** Mean error over 30 years of normalized dataset and the same number of normalized generated samples on 3 different pressure levels (1000, 500 and 100 hPa). The samples were horizontally transposed in order to have a Europe at the center of the images. Coastlines were added a posteriori for readability.




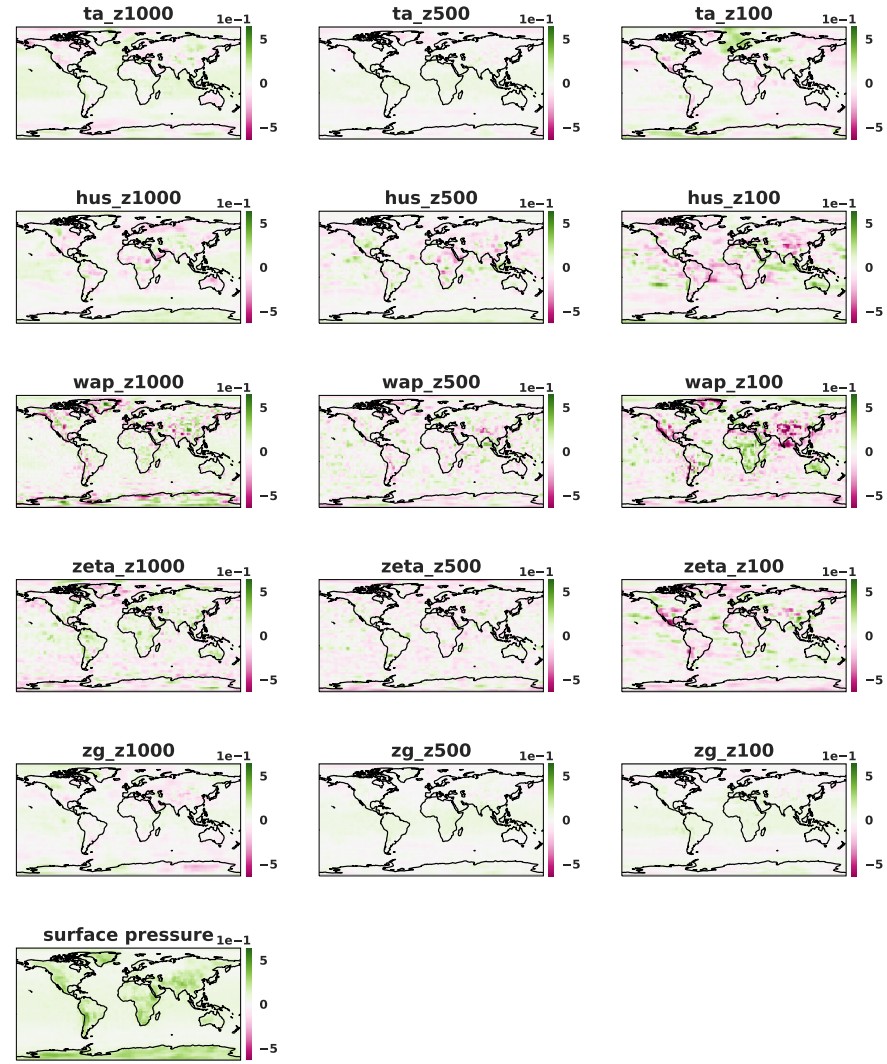

**Figure 11.** Standard deviation error over 30 years of normalized dataset and the same number of normalized generated samples on 3 different pressure levels (1000, 500 and 100 hPa). The samples were horizontally transposed in order to have a Europe at the center of the images. Coastlines were added a posteriori for readability.







**Figure 12.** Projection in subplans of five first components of SVD of 10800 normalized samples (30 years) from dataset (blue). Comparison with the projection in the same components of 10800 normalized generated sample (green). This was done on one field (geopotential height) at a pressure level z500 hPa.




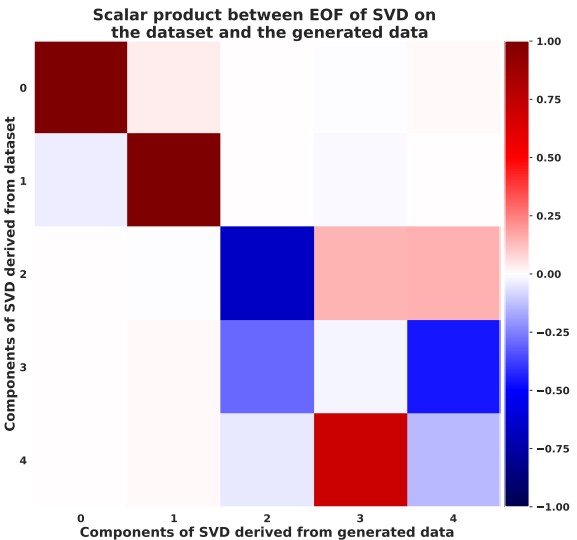

**Figure 13.** Scalar product of SVD components derived from dataset and generated data.

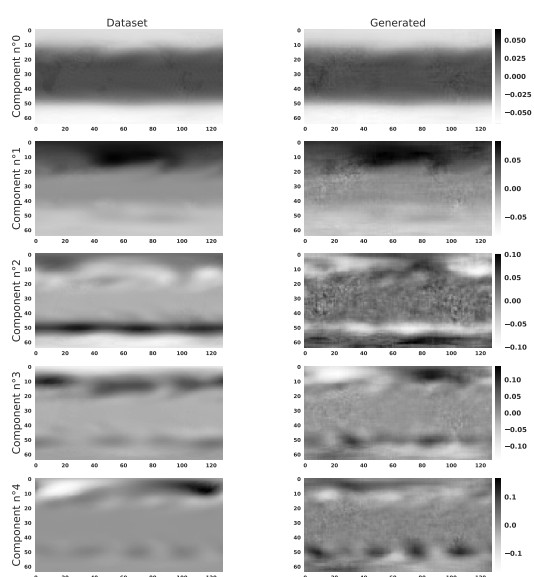

**Figure 14.** Spatial corresponding to principal components of SVDs applied to the dataset and the generated samples.





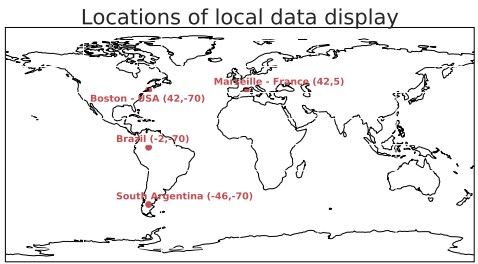

**Figure 15.** Location from where the temperature distribution are plotted in Fig. 16. The Wasserstein distance value associated for each plot was computed on normalized data.

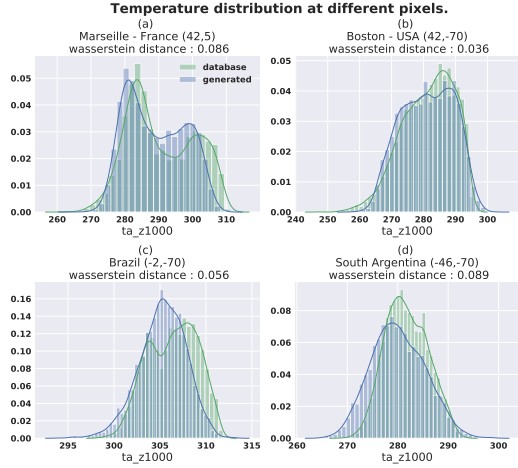

**Figure 16.** Temperature distribution at different locations for 5000 samples from dataset (green) and generated (blue).

mountainous areas are the most difficult to fit. As a way to better interpret this metric, Fig. 18 represents the distributions corresponding to the minimum and maximum values of the metric. The distribution of the Wasserstein distance can also be visualized grouped by pressure level and type of variable in Fig. 19. The $wap$ variable that represent the vertical velocity seems to be the one with the higher Wasserstein distance value.

### 3.3   Analysis of the atmospheric balances

The previous subsection has shown the ability of the generator to engender weather situations and climate similar to those of the simulated weather. However, geophysical fluids are featured by multivariate fields that present known balance relations. Among these balances the simplest ones are the geostrophic and the thermal wind balances (see *e.g.* Vallis (2006)). The next two sections assess the ability of the generator to reproduce the geostrophic and the thermal wind balances.





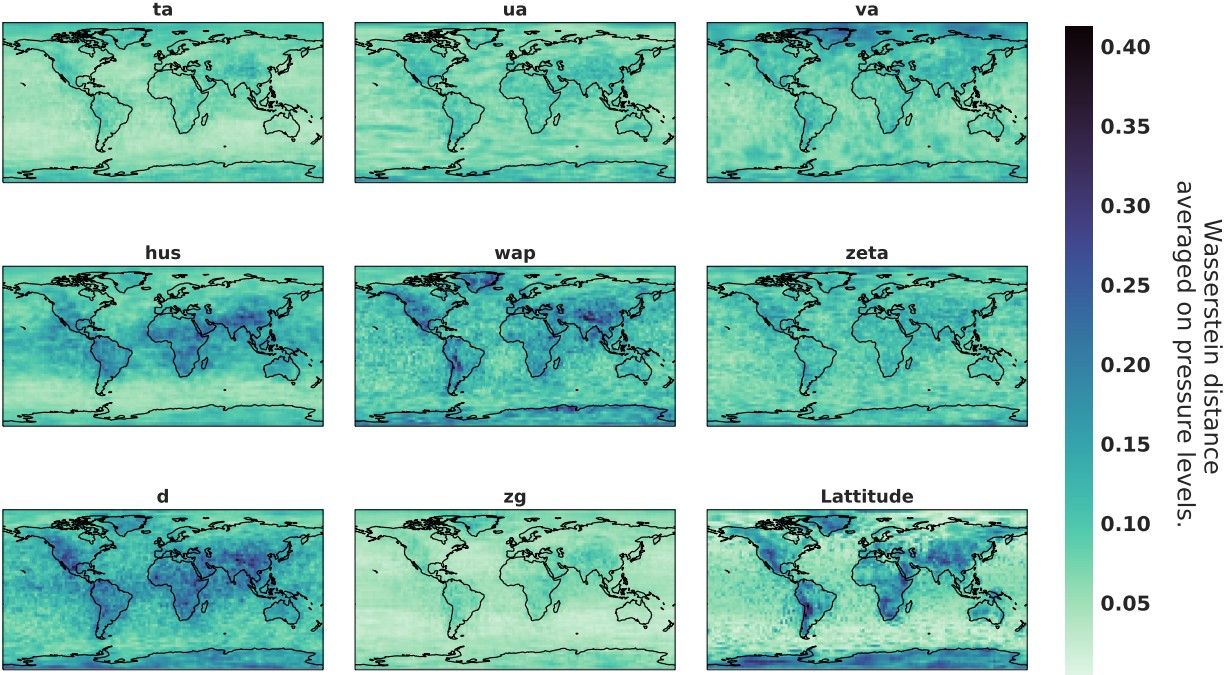

**Figure 17.** Wasserstein distance between 5 000 dataset and generated samples on each pixel and each channels.

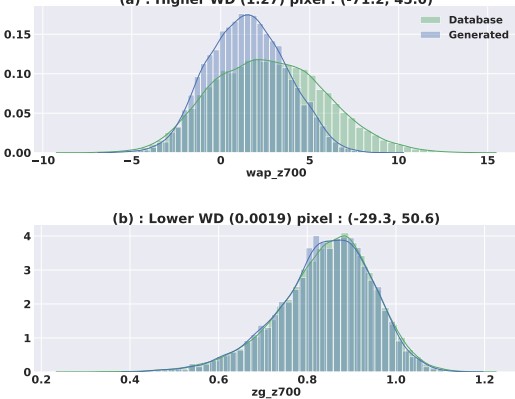

**Figure 18.** Distributions with the higher (top) and lower (bottom) Wasserstein distance computed on normalized data. The coordinates of corresponding pixels are respectively in latitude and longitude.


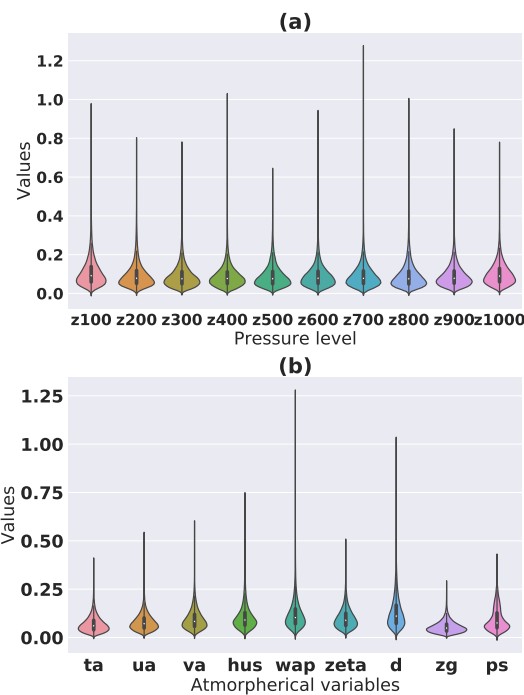

**Figure 19.** Wasserstein distance between 5 000 dataset and generated samples on each pixel grouped by pressure height (a) or variables (b).

### 3.3.1 Geostrophic balance

The geostrophic balance occurs at low Rossby number when the rotation dominates the nonlinear advection term. Two forces are in competition: the Coriolis force, $f\mathbf{k} \times \mathbf{u}$ where $\mathbf{k}$ denotes the unit vector normal to the horizontal, $f$ is the Coriolis parameter and $\mathbf{u}$ is the wind ; and the pressure term $-\nabla_p \Phi$ where $\Phi$ is the geopotential and where $\nabla_p$ denotes the horizontal gradient in pressure coordinate. Asymptotically, the Coriolis force, is then balanced by the pressure term which leads to the geostrophic wind

$$\mathbf{u}_g = \frac{1}{f}\mathbf{k} \times \nabla_p \Phi \qquad\qquad (15)$$

The geostrophic flow is parallel to the line of constant geopotential, and it is anti-clockwise (clockwise) round a region of low (high) geopotential. The magnitude of the geostrophic wind scales with the strength of the horizontal gradient of geopotential (Vallis, 2006, sec. 2.8.2, p 92).

     This asymptotic balance Eq. (15) is verified to within $10\%$ of error at mid-latitude, that is $\mathbf{u} = \mathbf{u}_g + \mathbf{u}_{ag}$ where the magnitude

of the ageostrophic wind, $\mathbf{u}_{ag}$, is less than $0.1$ of the magnitude of the real wind $\mathbf{u}$.

     Figure 20-(a) illustrates a particular boreal winter situation from the PLASIM dataset, focusing on the mid-latitude, and presenting a low area of geopotential on the southwest of Iceland. It appears that the wind is well approximated by the geostrophic



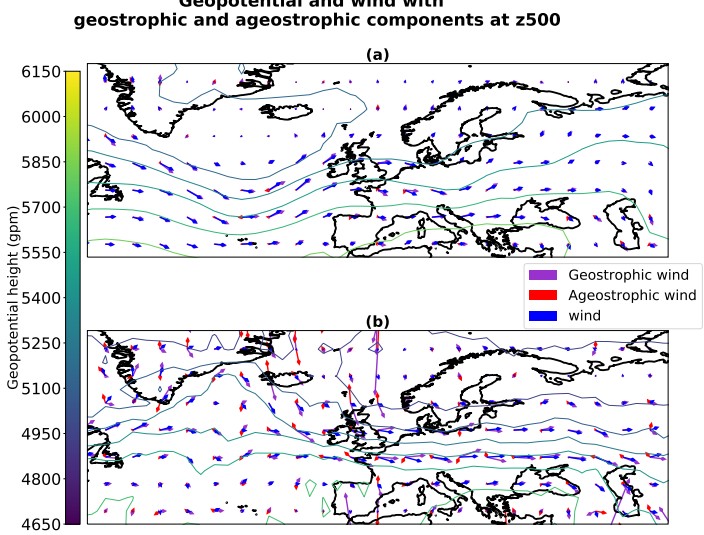

**Figure 20.** Geostrophic and ageostrophic wind derived from geopotential at 500 hPa. Situation taken from dataset (panel (a)) and generated (panel (b)).

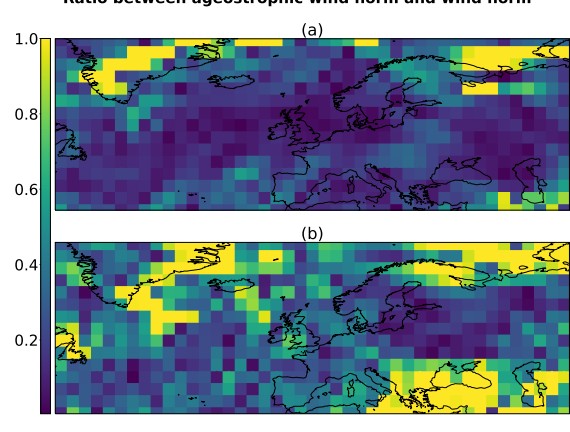

**Figure 21.** Relative error in norm between geostrophic wind and normal wind shown in Fig. 20.





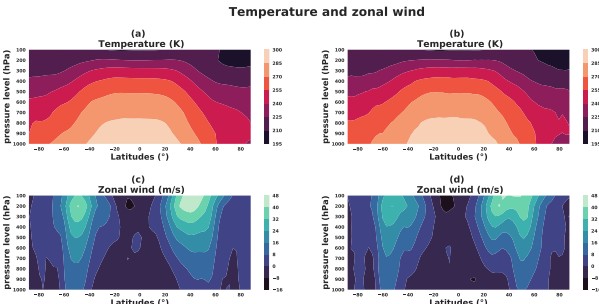

**Figure 22.** Temperature $(K)$ and zonal wind $(m/s)$ latitude zonals from a boreal winter situation: the thermal wind balance.: (a) sample from the dataset, (b) from a generated sample.

wind, which is quantitatively verified in Fig. 21-(a) that shows the norm of the ageostrophic wind normalized by the norm of the wind (that is the relative error when approximating the wind by the geostrophic wind): the order of magnitude of the error

is around 20%. Properties of the geostrophic flow are visible, with an anti-clockwise flow around the low geopotential. The wind is maximum where the horizontal gradient of geopotential is maximum while its change of direction follows the trough.

A similar behavior can be observed in Fig. 20-(b) which illustrates a weather situation selected from the render by the generator of some samples in the latent space, so to represent a boreal winter situation. This time, a low geopotential is found on the north of the Europe. While the geopotential field is noisy (it is less smooth than in panel (a)), the wind is again found

nearly geostrophic verifying the geostrophic flow properties to within an error of 35%. (see Fig. 21-(b)). The geopotential and wind fields were projected on the solved dynamic truncation in order to remove the subgrid component due to the noise in the output of the generator. Despite the truncation the geostrophic approximation seems to not be respected everywhere and could be a quantitative metric to monitor in order to improve our method.

We find that weather situations generated from samples in the latent space reproduce the geostrophic balance at an order

of approximation that is similar to the one of the real data set. This means that the generator is able to produce the realistic multivariate link between the wind and the geopotential. This property is essential in operational weather forecasting *e.g.* in producing balanced fields in ensemble Kalman filter.

### 3.3.2   Thermal wind balance

The thermal wind balance arises by combining the geostrophic wind Eq. (15) and the hydrostatic approximations, $\frac{\partial \Phi}{\partial p} = -\frac{1}{\rho}$

where $\rho$ is the density (Vallis, 2006, sec. 2.8.4, p95): Taking the derivative of Eq. (15) with respect to the pressure $p$ makes appear the hydrostatic approximation so that the vertical derivative of the geostrophic wind writes

$$\frac{\partial \Phi}{\partial p} = -\frac{R}{pf}\mathbf{k} \times \nabla_p T, \tag{16}$$



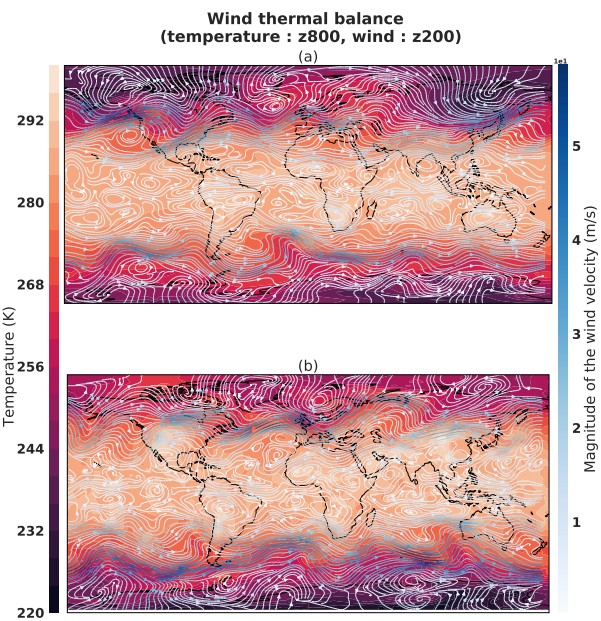

**Figure 23.** Thermal wind balance from the boreal winter situation shown in Fig. 22: (a) sample from the dataset, (b) sample generated by the generator. The temperature $(K)$ is from pressure level 800 hPa and the wind $(m/s)$ from 200 hPa.

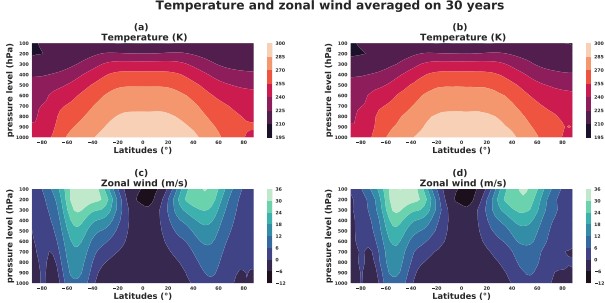

**Figure 24.** Temperature $(K)$ and zonal wind $(m/s)$ latitude zonals averaged on 30 years subsample



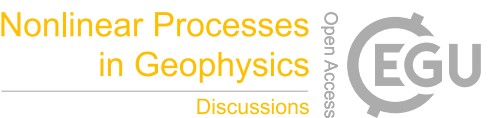

where the ideal gas equation, $p = \rho R T$, has been used. Eq. (16) is the thermal wind balance that relates the vertical shear of the horizontal wind to the horizontal gradient of temperature. In particular, when the temperature falls in the poleward direction,

the thermal wind balance predicts an eastward wind that increase with height.

The Fig. 22-(a & b) shows the vertical cross-section of the zonal average of temperature and of the zonal wind for a particular weather situation in the dataset, corresponding to a boreal winter situation of the same weather situation represented in Fig. 22 : the temperature is higher in the Southern than in the Northern Hemisphere with a strong horizontal gradient of temperature in latitude ranges $[-80°, -40°]$ and $[40°, 80°]$. At the vertical of the horizontal gradient of temperature, the wind is eastward and

increases with the height: this illustrates the thermal wind balance which produces strong curled jet at the vertical of strong horizontal gradient of temperature as shown in Fig. 23-(a) that illustrates, for the same weather situation, the temperature at the bottom (800hPa) with the horizontal wind at the top (200hPa) of the troposphere.

The same illustrations are shown in Fig. 22-(c & d) when considering a generated situation, selected to correspond to a boreal winter situation: the characteristic related to the thermal wind balance as observed before are found again. It results

that the generator is able to render a weather situation that reproduce the thermal wind balance. Moreover, Fig. 24 shows the thermal wind balance averaged on 30 years for the dataset (panel a) and generations (panel b) both are very similar.

This section has shown the ability of the generator to reproduce some important balances present in the atmosphere. In particular, the generator is able to produce mid-latitude cyclones whose velocity field is in accordance with the geostrophic balance. It would be interesting to consider other diagnostics to complete the evaluation of the realism of the generated states.

Note that adding advanced diagnostic fields in the output of the generator could be investigated to improve the realism.

### 3.4   Exploration of the latent space structure and its connection to the climate

An exploratory study was done on the property of the latent space and its consequence in the climate space in regard to climate domain problematic. If the generator is perfectly trained, then each sample generated with it should represent a typical weather situation. It is hard to figure out what is the attractor of the climate. However, the geometry of the Gaussian in high dimension

being known, it is easy to characterize the climate in the latent space.

#### 3.4.1   Geometry of the normal distribution

For a normal law in the high dimension space $Z = \mathbb{R}^m$ *i.e.* with $m$ larger than 10, the distribution of the samples are all located in a spherical shell of radius $\sqrt{m}$ and of thickness of order $\frac{1}{\sqrt{2}}$ (see *e.g.* Pannekoucke et al. (2016)). Because the covariance matrix $\mathbf{I}_m$ is a diagonal of constant variance, no direction of $\mathbb{R}^m$ is privileged leading to an isotropic distribution of the direction

of the sampled vectors: their unit directions uniformly covers the unit sphere. Another property is that the angle formed by two sampled vectors is approximately of magnitude $\frac{\pi}{2}$: two random samples are orthogonal. These are simple consequences of the central limit theorem which predict, for instance, that the distance of a sample to the center of the sphere is asymptotically the Gaussian $\mathcal{N}\left(\sqrt{m}, \frac{1}{2}\right)$.

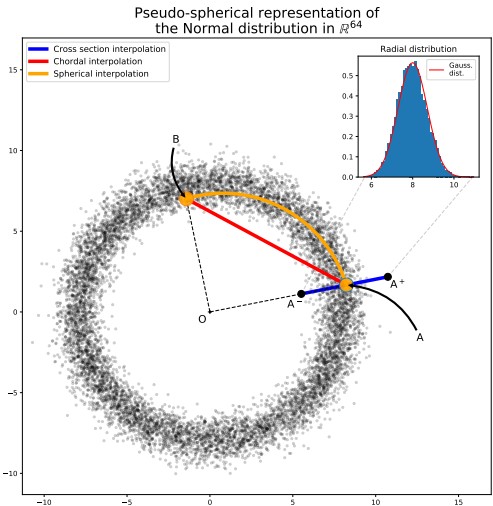

**Figure 25.** Pseudo-spherical metaphorical representation of $10000$ samples of the normal distribution in $\mathbb{R}^m$ with $m = 64$ and the distribution of the distance of samples to the center of the spherical shell. For a sample $A$, $A^{\pm}$ denote two extremes situations along the direction of $A$. Any second sample $B$, typical of the distribution, appears orthogonal to $A$. The inset figure represents the radial distribution, compared with the asymptotic CLT Gaussian distribution $\mathcal{N}\left(\sqrt{m}, \frac{1}{2}\right)$ (thin red curve).

Considering these properties, one can introduce a 2 dimension pseudo-representation which preserves the isotropy of the

distribution as well as the distribution to the origin: a random sample vector $\mathbf{x} = (x_1, x_2, \cdots, x_m)$ in $\mathbb{R}^m$ is represented by the projection $P_2(\mathbf{x}) = \|\mathbf{x}\| \frac{1}{\sqrt{x_1^2 + x_2^2}}(x_1, x_2)$ where $\|\cdot\|$ stands for the Euclidian norm in $\mathbb{R}^m$.

Fig. 25 illustrates this low-dimensional representation of an ensemble of $10000$ samples of the normal law in dimension $m = 64$. For instance, points $A$ and $B$ represent two independent samples: their distance to the origin is closed to $\sqrt{m} = 8$, and their angle is closed to $\frac{\pi}{2}$. While $m = 64$ can be considered as a very small dimension, it appears that the distribution of the

point's distance to the origin is well fit by the Gaussian $\mathcal{N}\left(\sqrt{64}, \frac{1}{2}\right)$ (see inset figure in Fig. 25). Hence, it results that for this dimension, the interpretation of a Gaussian distribution as spherical shell applies, with interesting consequences for extremes or typical states. A typical sample of this normal law is a point near the sphere of radius $\sqrt{64}$ while an extreme sample has a norm lying in the tails of the distribution $\mathcal{N}\left(\sqrt{64}, \frac{1}{2}\right)$.

This suggests evaluating whether the extremes of the latent space correspond to those of the meteorological space.

**3.4.2 Connection between extremes in the latent and the physical spaces**

Knowing what are the extremes in the latent space, might be helpful to determine what are the extremes of the climate, at least to determine what are extreme situations closed to a given state.





For any sample in the latent space, say point $A$, we can construct the point on the sphere $\sqrt{m}$ along the same direction of $A$, $\bar{A}$, which can be considered as the most likely typical state near $A$. Along the same direction of $A$, we can also construct the extreme situations $A^{\pm}$ whose distances to the origin, $\sqrt{m} \pm \frac{3}{\sqrt{2}}$, lie respectively in the left and in the right tails of the Gaussian distribution $\mathcal{N}\left(\sqrt{m}, \frac{1}{2}\right)$.

Fig. 26 represents the weather situation generated from a randomly drawn latent vector from a 64-dimensional Gaussian $\mathcal{N}(0,1)$ sample $A$ (panel a). The panel A represents a latent vector with an Euclidian norm equals to 7.69, close to the mean of the radial distribution of the hypersphere mentioned in the Sect. 3.4.1. In the climate space this sample shows a meteorological object above the Northern Europe in shape of a geopotential minimum which can be interpreted as a storm. This sample is the same that the one represented in the panel (b) of Fig. 20, 22, 23.

The most likely typical state $\bar{A}$ (panel b) is the radial projection of the latent vector $A$ on the mean of radial distribution, thus its Euclidian norm is equal to 8. Because the sample $A$ have a norm close to the sample $B$, the weather situations are very similar on the geopotential height at z1000. This is an expected effect because by construction of the generator the input space is continuous so two points in the latent space must be similar. Extreme situation $A^{\pm}$ along the direction of $A$ are represented in panels (c) and (d). Both panels shows clear differences on the geopotential height. First the panel (c) shows a decrease in the storm located above Northern Europe, same effect is visible at the south of South-America. However, the weather situation is very similar to the panel (a). On the contrary panel (d) represents a deeper geopotential height minimum at pre-existing storm of sample $A$. Thus, the Fig. 26 seems to show a certain structure of the latent space generator where the radial direction could represent the strength of the meteorological objects such as storms above Europe for example. It could be explained by the fact that the generator aims to map a distribution (64-dimensional Gaussian in the latent space) to another (weather distribution in the PLASIM physical space). Rare events exist in the latent space on the tails of the Gaussian distribution potentially extreme weather situations. One of the way to do a such mapping is to use the radial direction to represent high or low probability states of the climate. An important conclusion is that, for a given situation, the most-likely state and the extremes are interesting physical states. This could open new possibilities to study extreme situation close to a given one, which is an important topic *e.g.* for insurance or to improve the study of high weather impact in ensemble forecasting.

The link of the animation of such interpolation is available on the GitHub[1] of the project.

### 3.4.3 Interpolation in the latent space

Even if there are no dynamics in the latent space, which makes it impossible to construct a prediction from this space, we can consider how to interpolate two latent states. A naive answer is to compute the linear interpolation between two samples of the latent space $A$ and $B$,

$$M_t = G\left((1-t)A + tB\right) \tag{17}$$

---

[1]https://github.com/Cam-B04/...NOTDEFINED

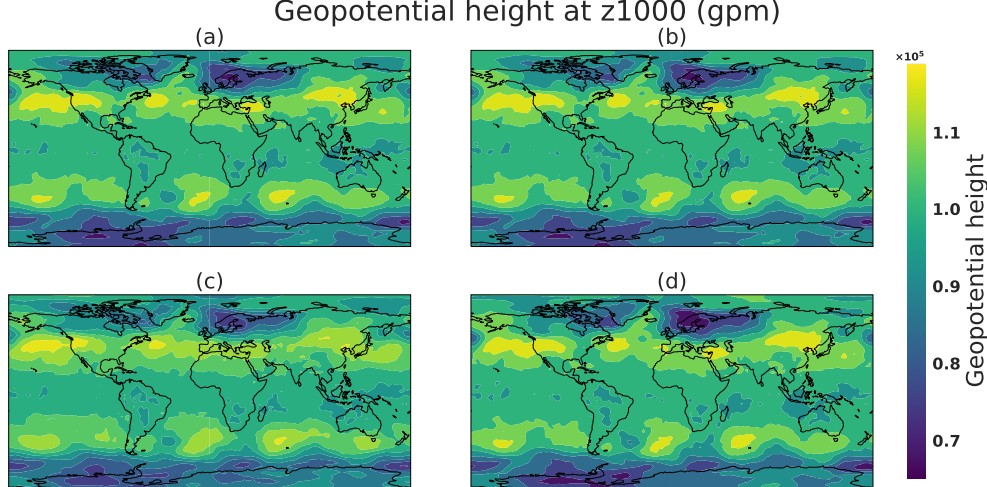

**Figure 26.** Generations obtained by radial interpolation in the latent space. (a) is the image corresponding to a randomly drawn latent vector $A$ (2-norm : 7.69), (b) is its projection on the mean of the same direction $\bar{A}$ (2-norm : 8.0), (c) and (d) are the projection on respectively inferior $A^-$ (2-norm : 5.87) and superior $A^+$ (2-norm : 10.12) 1% quantile (see Fig. 25).

which results to the red chordal illustrated in Fig. 25. The chordal interpretation highlight a major drawback of the linear interpolation: middle points of the chordal are extremes, these intermediate points should not correspond to typical (or even physically realizable) weather situations.

So to preserve the likelihood of the interpolated weather situations, it is better to introduce a spherical interpolation. This kind of interpolation has also been used in image processing where *e.g.* White (2016) use the formula

$$M_t = G\left(\frac{\sin((1-t)\theta)}{\sin\theta}A + \frac{\sin(t\theta)}{\sin\theta}B\right), \tag{18}$$

where $\theta$ is the angle $\widehat{A,B}$, and for $t \in [0,1]$ such as $M_0 = G(A)$ and $M_1 = G(B)$.

This interpolation will connect point $A$ to $B$ within the spherical shell of typical states, as illustrated by the orange curve line in Fig. 25. The Fig. 27 shows snapshots of the climate generated from a spherical interpolation in the latent space between the sample $A$ and another random sample B. For the sake of comparison the Fig. 28 and 29 are respectively snapshots of a linear interpolation in the latent space and the image space.

The objective of this experience is to be able to produce realistic intermediate states. This can be visible on the Fig. 27 where the storm above Europe is emerging by first a smaller minimum in geopotential height that increases in size. Whereas on both linear interpolations, in the latent and image space, the storm appears first as a long and thin geopotential minimum then broaden on the latitude direction. Such property can be helpful in the context of fluid dynamics for initial and boundary conditions of a local area model to avoid error correlated with user defined parameters such as in Lateral Boundary conditions Davies (2014). An interesting generator property would be to be able to choose some characteristics of the generated climate


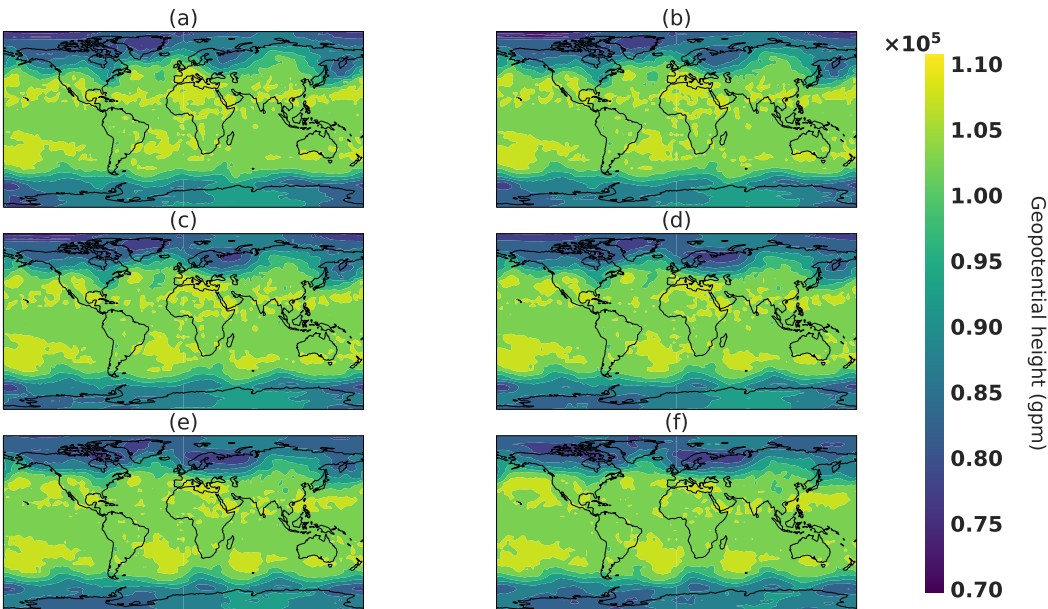

**Figure 27.** Spherical interpolation snapshots. Respectively the panels (a), (b), (c), (d), (e) and (f) correspond to a value of t in Eq. (18) of 0, 0.2, 0.4, 0.6, 0.8, 1.

such as meteorological objects at certain locations. In the next section, an experiment is conducted to see if it is possible to change the location of such meteorological objects.

### 3.4.4   Coherent structure perturbation from the latent space

In this section, the goal is to study the difference between two climate states coming from close latent points. In this experiment, the sample $G(A)$ will be the reference climate state, and we added noise to $A$ such as $A = A + \epsilon_i$ with $\epsilon_i$ taken from $\mathcal{N}(0, 0.1)$.

Figure 30 shows the different climate states corresponding to $G(A)$ and $G(A+\epsilon_i)$ in the first column and the difference with the reference in the climate states $G(A) - G(A+\epsilon_i)$ in the second column. The second column shows dipoles that represent the movement of meteorological structures for example in the South-America area of panel (d). We remarked that the perturbation of one latent vector is translated in the climate state by a dipole creation when the difference is done between the reference and the perturbed version. This shows the possibility to move meteorological object by remaining on the manifold of the realistic

climate state. This is an interesting asset for the climate domain where it is complicated to interpolate between two states where a storm is at two different locations as mentioned in Hergenrother et al. (2002). The WGAN could be a way to propose realistic intermediate states.




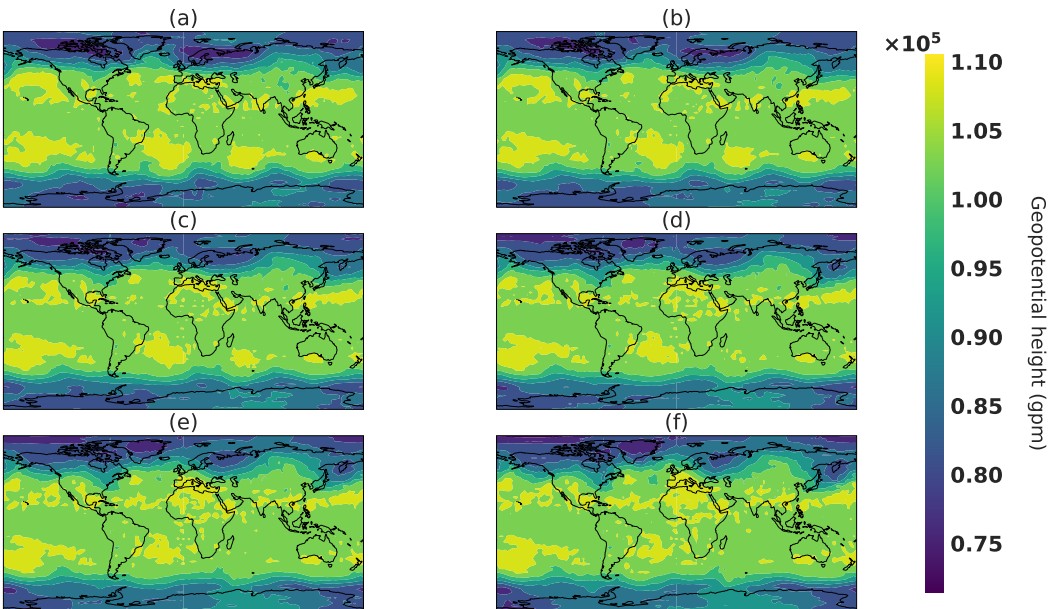

**Figure 28.** Linear interpolation in the latent space interpolation snapshots. Respectively the panels (a), (b), (c), (d), (e) and (f) correspond to a value of t in Eq. (17) of 0, 0.2, 0.4, 0.6, 0.8, 1.

## 4  Conclusion

Our study shows that it is possible to map the climate distribution output of a GCM to a much simpler low-dimensional distri-

bution using a highly non-linear neural network based generator. It also proposes ways to assess the quality of the generator by evaluating statistical quantities as well as the respect of physical balance properties.

In this article, a weather generator based on the WGAN method able to produce realistic states of the atmosphere was created. Metrics such as SVD principal components' comparison, Wasserstein distance on pixel value distribution and mean and standard deviation comparison were used in order to be compared to other future proposed methods.

A comparison of the atmospheric balance was realized between samples and averaged over 30 years of data, showing promising results. Coherence between variables, as well as spatial coherence were also shown to be promising.

Interesting properties of such generator were discussed with regard to possible applications in insurance, weather simulation and data assimilation. The generator is able to generate intermediate realistic climate states with coherent structures, interpolate between 2 defined states with other plausible states, and create realistic perturbations around a climate state, all at a low-

computational cost compared to a GCM.


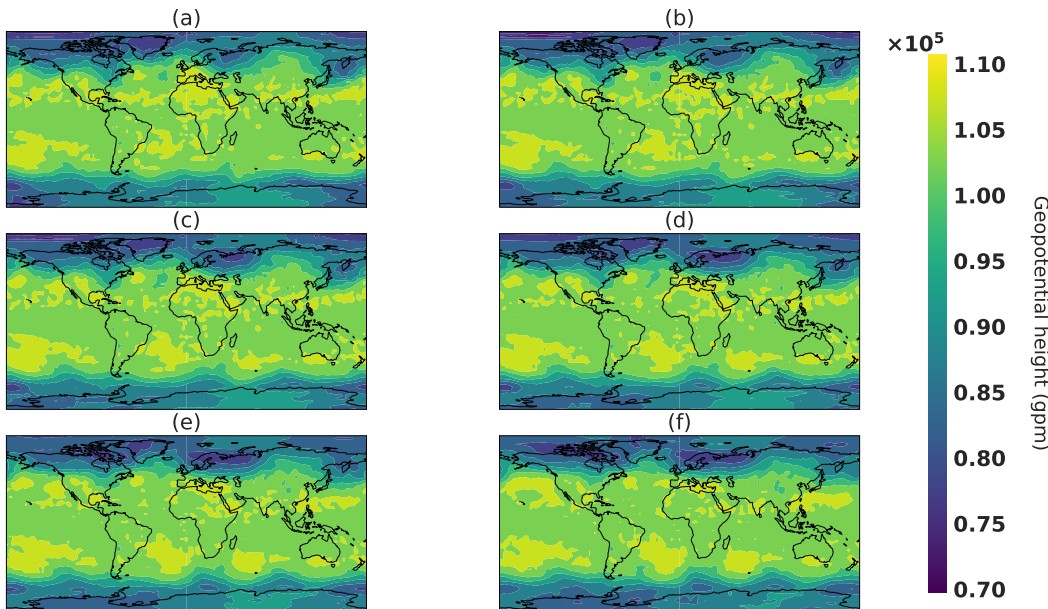

**Figure 29.** Linear interpolation in the image space. Respectively the panels (a), (b), (c), (d), (e) and (f) correspond to a value of t in $M_t = (1-t)G(A) + tG(B)$ of 0, 0.2, 0.4, 0.6, 0.8, 1.

A study was also done on the interpretability of the latent space and the connections between the extreme events in the data space and the latent space. It highlighted the radial direction as the direction of the intensity of climate events.

Our results highlight the ability of the method to handle the mapping of a high-dimensional distribution onto a multivariate Gaussian. We believe this is an important step that opens many opportunities for climate data exploration. Some extensions of this work as well as potential application are highlighted in the following.

First, the WGAN could be conditioned by the season or by the day in the year. Such conditioning would give access to other quantitative methods to assess the quality of the weather generator. It would be also an important step towards the application in risk assessment area for example.

Optimization can be done to find specific states in the latent space by defining an objective function such as Euclidian distance in the climate space. The network gradient with respect to its inputs being accessible, direct minimization can be used to find climate states that fit observations in data assimilation problems. More advanced strategies, such as training a separate inference network (Chan and Elsheikh, 2019) are also possible to apply Bayes' rule without using a particle filter.

A more sophisticated dataset could be used, such as a true climate reanalysis, to see the effect of the dataset complexity on the method's performance. The optimization of the network's architecture and a sensitivity study on the hyper parameters such

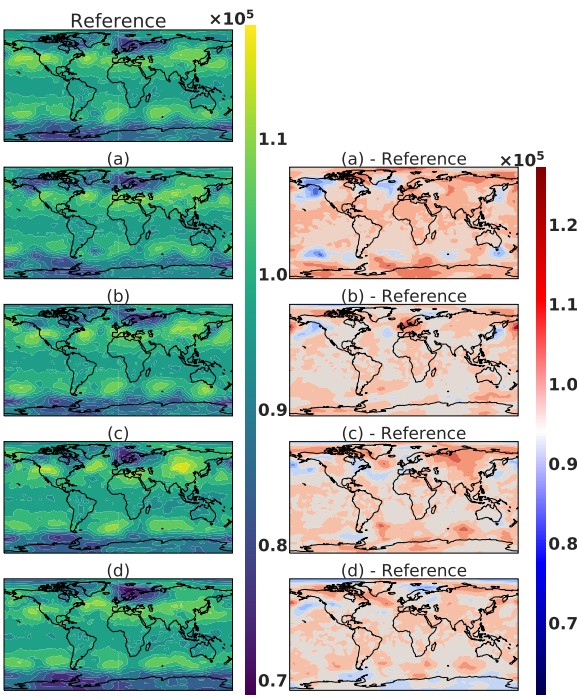

**Figure 30.** Geopotential height corresponding to : first column $G(A)$ and $G(A+\epsilon_i)$ and second column $G(A) - G(A+\epsilon_i)$

as the dimension of the latent space for example would be useful. Moreover, it would be interesting to see if it is possible to take advantage of the GAN trained in PLASIM to facilitate the training of a GAN on the reanalysis.

     The structure of the latent space and its interpretability is also a critical way to exploit the specificities of the method. The opportunity to find similar climate states with extreme events is also something not possible with other weather generators and could have lots of application for risk assessment applications.

The definition of additional metrics to assess the quality of the generator should be the main focus following this study to identify improvement of the method and facilitate the participation from diverse researcher communities.

     Finally, we could consider restarting the GCM from a generated state to assess how well-balanced the generated fields are, which could have important implications in data assimilation methods.

     The study is a first step towards deep learning weather generation, while many challenges remain to be solved, it shows

several potential applications of GANs to improve the effectiveness of current approaches.





*Code availability.* The code and the weights of the trained neural network is available at the following GitHub repository in v0.1 : https://github.com/Cam-B04/Producing-realistic-climate-data-with-GANs.git. The repository is associated with the following DOI : 10.5281/zenodo.4442450

*Data availability.* The dataset used is available on demand. The github repository explains how to recreate it from a PLASIM simulation.

*Author contributions.* The authors contribute to the design of the neural network architecture and the experiments. C. Besombes implemented
the neural network architecture, performed the PLASIM simulation and trained the WGAN. The analysis of the results has been made by the authors.

*Competing interests.* The authors declare that they have no conflict of interest.

*Acknowledgements.* This research paper was written during a thesis in partnership with Total. We would like to thank Philippe Berthet, Anahita Abadpour, Daniel Busby and Tatiana Chugunova for their support in the application of our method in different fields of geosciences.
We would like to thank Rabeb Selmi for her help and for sharing her expertise.



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
