# Peer review of "Producing realistic climate data with GANs"

_Nonlinear Processes in Geophysics, 2021_

## Author Response (AR2)

First of all, we would like to thank the referee for their review of our paper and for giving us the opportunity to improve it.

The structure of the response will be the detailed answers to the questions raised by the referee and then the list of the changes made compared to the preprint version :

**Answers to the referee's questions :**

RC1-C1.
How tolerant are the wrap padding layers to discontinuity?
We have added in supplementary material Fig. 6 that is a plot of the value at z100 and z500 levels along the longitude for a given sample where the dashed line shows the x index corresponding to the sides of the image. It can be observed, in this sample, that no discontinuity can be seen. This test was reproduced for other samples where no discontinuities were detected as well.
We also note that the original simulator, PLASIM conserves continuity across the boundary intrinsically. It was also added in the text of the article that *"PLASIM simulator is a spectral model run on a gaussian grid that consequently enforces the periodic boundary condition." (line 174, page 7)*

To which extent, the padding process may affect the training ?
The discontinuities can be seen as unwanted spatial patterns. In the early training phase, the generator will produce noisy images. During training iterations, the generator will remove unwanted spatial patterns. The wrap padding layer can be seen as a way to underline those unwanted spatial patterns (here discontinuities) between points in the images that are far from one another. Thus, authors argue that the padding is necessary – but not that the effect is insignificant. The computational cost is too important to repeat the analysis for a range of padding values.

RC1-C2 :
By looking at the inputs/outputs in figures 1 to 5, I assume that some convolutional blocks have pooling/ upsampling layers and others do not. Could you explain the reason for that and present these blocks differently in the figures?
The upsampling and pooling layers are here to respectively increase and decrease the size of the image until it reached the desired output size. Once the image has the right size it can be useful to add convolutional block in order to increase the capacity of the network but these additional convolutional layers will not need to modify the image shape.
In our case, the pooling layer is a strided convolution that can be seen as a learnable pooling layer (see Fig 1 to 5 in supplementary material). The same could be done for upsampling layer, where it can be replaced by a strided transposed convolution. In the sake of simplicity, it was decided to keep upsampling layer for our case.
Added sentence in the article : *"At each strided convolutional block, $s=2$ in Fig. 3 the image size is divided by a factor 2. It is equivalent to a learnable pooling layer that can increase the result (Springenberg 2014)." (line 190, page 7)*

RC1-C4 :

The stopping criteria of the training is a maximum number of iterations. Could this condition be combined with the convergence of the loss function, for example loss function derivative very small or training set and validation set having almost equal loss ?

In theory, because the wasserstein distance, and its derivative, has a mathematical meaning, it could be used as a stopping criteria. But in application it is not yet proven that the WGAN will converge the Nash Equilibrium in a finite number of iterations. Moreover, the instability of the GAN training is a difficulty for using the derivative of the loss function. Thus, to define a threshold under which the loss function should be under to stop the training, it would be necessary to first define metrics related to the data domain (as we have done in our study). The concept of training and validation set for the generator is not pertinent because it is a non-supervised training. However, it could be useful to perform such dataset split to evaluate the discriminator performance on images it has never seen.

Remark about this question was added in the manuscript :

*"Although it is possible to use the loss of the critic as convergence criteria because the Wasserstein loss is used and has a mathematical meaning such as the distance between synthetic and real data distributions and should converge to 0. But WGAN-GP is not yet proven to be locally convergent under proper conditions (Nagarajan 2017), the consequence is that it can cycle around equilibrium points and never reach a local equilibrium. Condition on loss derivative is also difficult because of the instability of the GAN training procedure. Consequently, a quality check using metrics adapted to the domain on which the GAN is applied is still necessary."* (line 221, page 13)

RC2-C1 :

I feel that the manuscript can attract more readers by providing more information on the background of the study. How about adding more references about weather generators in the introduction?

A paragraph was added in the introduction that presents the different data-driven methods applied to numerical weather prediction :

*"Data driven approaches and numerical weather prediction are two domains that share important similarities, Watson-Parris (2021) explains that both domains use the same methods to answer different questions. This study and Boukabara et al. (2019) also show that numerical weather prediction contains lots of interesting challenges that could be tackled by machine learning methods. It clarifies the growing literature about data driven techniques applied to weather prediction. Scher (2018) used variational autoencoders to generate the dynamics of a simple general circulation model conditioned on a weather state. Weyn et al. (2019) trained a convolutional neural network (CNN) on gridded reanalysis data in order to generate 500-hPa geopotential height fields at forecast lead times up to 3 days. Lagerquist et al. (2019) developed a CNN to identify cold and warm front and post-processing method to convert probability grids to objects. Weyn et al. (2020) built a CNN able to forecast some basic atmospheric variables using a cubed-sphere remapping in order to alleviate the task of the CNN and impose simply boundary conditions."* (line 33, page 2)

RC2-C2.1 :

The authors evaluated the spatial distribution of the variables simulated by the generator and showed that the generator's performance is generally good. This is good to evaluate the generator but lacks thoroughness.

The authors should evaluate how well the inter-variable relationships are simulated by the generator ?

We agree with the referee, some of the preliminary studies was done such as verification of the geostrophic balance (Fig 21.), Thermal wind balance (Fig 22, 23, 24.). It would be worth to continue this analysis but it is beyond the scope of our study which is a proof of concept. We have added to the discussion a consideration of the inter-variable relationships that should be studied in further analysis: Omega equation, other inter-variables balances.

RC2-C2.2 : It is also better to investigate the reproducibility of the vertical structure.
? We have, to some degree illustrated that the approach reproduces some elements of the vertical structure such as Wasserstein distance of variable distributions generated by the neural network with respect to altitude (Fig. 19) and temperature and zonal wind (Fig. 22 & 24) and it would be worth to continue this analysis but it is beyond the scope of our study which is a proof of concept.
Added sentence in the manuscript : *"Authors emphasize that it is necessary to conduct more analysis of the weather situations outputted by the generator which is beyond the scope of this study. For example it would be interesting to assess if other inter-variable balances are present such as the $\omega$-equation, vertical structures etc."(line 364, page 29)*

RC2-C4 : Similarly, is it possible to give the generator SST fields (or any other boundary conditions (e.g. GHGs, orbital parameters, solar, etc.)) as inputs? For the generator to be used for applications you mentioned in the manuscript (e.g. risk assessment, data assimilation), such information should be included as the inputs.
Using input optimization at inference could be used to condition generation for specific properties in the generation. It is also possible to condition the generations to a specific date in the annual cycle with slight modifications in the network architecture. One could think to condition the output of the generator by a forcing field in input such as forcing field like SST fields for data assimilation application, which should be possible but with more important modifications of the network architecture and a possible impact on the speed of the training procedure.
Possible ways of conditioning our generator (by date, or input forcing fields) were added and the estimation of the changes necessary to be done to the network architecture : *"More advanced strategies, such as training a separate inference network (Chan 2019.) are also possible to apply Bayes' rule without using a particle filter. It is also possible to condition the generations to a specific date in the annual cycle with slight modifications in the network architecture. One could think to condition the output of the generator by a forcing field in input such as forcing field like SST fields for data assimilation application, which should be possible but with more important modifications of the network architecture and a possible impact on the speed of the training procedure." (line 473, page 35)*

RC2-C5 : The most important advantage to use weather generators is, I believe, its low computational cost. Therefore, the authors should show how fast the weather generator is compared to the GCM used as the training data.
The GCM simulation was done in 50 minutes on 16 processors and the generator generated 36500 samples (daily sample for 100 years simulation) in 3 minutes on an NVIDIA V100. The comparison of the computational cost between the GCM and the generator was added to the manuscript : *"The first required property for a weather generator is a low computational cost compared to the GCM that produced the data. Here the simulation with the GCM PLASIM took $50$ minutes for a $100$-year simulation in parallel on $16$ processors, whereas the generator took $3$ minutes to generate $36500$ samples equivalent to a $100$-year simulation on an NVIDIA Tesla V-100." (line 267, page 15)*

**Minor changes :**

A paragraph was added in the introduction that presents the different data-driven methods applied to numerical weather prediction :
*"Data driven approaches and numerical weather prediction are two domains that share important similarities, Watson-Parris (2021) explains that both domains use the same methods to answer different questions. This study and Boukabara et al. (2019) also show that numerical weather prediction contains lots of interesting challenges that could be tackled by machine learning methods. It clarifies the growing literature about data driven techniques applied to weather prediction. Scher (2018) used variational autoencoders to generate the dynamics of a simple general circulation model conditioned on a weather state. Weyn et al. (2019) trained a convolutional neural network (CNN) on gridded reanalysis data in order to generate 500-hPa geopotential height fields at forecast lead times up to 3 days. Lagerquist et al. (2019) developed a CNN to identify cold and warm front and post-processing method to convert probability grids to objects. Weyn et al. (2020) built a CNN able to forecast some basic atmospheric variables using a cubed-sphere remapping in order to alleviate the task of the CNN and impose simply boundary conditions."* (line 33, page 2)

The formatting of the equations 5, 10, 13 was reviewed.

Figure 1 modified in order to specify the layers where strided convolution were used. (see Fig. 1 in supplementary material)

Figure 2 and 3 : Captions were modified to differentiate convolutional and identity blocks. (see Fig. 2 and 3 in supplementary material)

Figure 3 was modified to introduce the parameter *s* representing the stride value of the convolution that change between different convolutional blocks in the critic architecture. (see Fig. 3 in supplementary material)

Figure 4 : The mention of the upsampling layers, when used, were added on the convolutional blocks. (see Fig. 4 in supplementary material)

Figure 5 : Caption modified to precise that upsampling layer is not in all convolutional blocks. (see Fig. 5 in supplementary material)

The merging of figure 10 and 11 was not done due to a loss of readibility of the figures.

Figure 14 was removed because of the redundancy of the information it shows.

Figure 22, 23 and 25 : Descriptions of the panels (a), (b)... were added.
> Figure 22 : *"Temperature $(K)$ and zonal wind $(m/s)$ latitude zonals from a boreal winter situation : the thermal wind balance. Left panels correspond to a situation taken from dataset (a) zonal temperature, (c) zonal wind, right panels correspond to a situation taken from generator (b) zonal temperature and (d) zonal wind."*

Figure 23 : *"Thermal wind balance from the boreal winter situation shown in Fig. 22 : (a) sample from the dataset, (b) sample generated by the generator. The temperature $(K)$ is from pressure level 800 hPa and the wind $(m/s)$ from 200 hPa."*

Figure 24 : *"Temperature $(K)$ and zonal wind $(m/s)$ latitude zonals averaged on 30 years subsample. Left panels correspond to a situation taken from dataset (a) zonal temperature, (c) zonal wind, right panels correspond to a situation taken from generator (b) zonal temperature and (d) zonal wind."*

Figure 30 : The equation was referenced instead of described in the caption.

**Supplementary material for the answer to the reviewers :**

[Figure]

*Figure 1 - Critic architecture*

[Figure]

*Figure 2 Residual identity block for the critic*

[Figure]

*Figure 3 Residual convolutional block for the critic. If s is different from 1 it is referenced as strided convolutional block in Fig.*

[Figure]

*Figure 4 Generator Architecture*

[Figure]

*Figure 5 Residual convolutional block for the generator. The upsampling layer can be removed if not necessary and is mentioned when used in Fig. 4*

[Figure]

*Figure 6 Variables of 1 sample from the dataset and 1 from the generator along the longitude for 2 levels z100 and z500. The dashed line represents the domain boundary where the periodic boundary condition should be enforced.*